# STYLEBENCH: EVALUATING THINKING STYLES IN LARGE LANGUAGE MODELS

## ABSTRACT

The effectiveness of Large Language Models (LLMs) is heavily influenced by the reasoning strategies, or *styles of thought*, employed in their prompts. However, the interplay between these reasoning styles, model architecture, and task type remains poorly understood. To address this, we introduce StyleBench, a comprehensive benchmark for systematically evaluating reasoning styles across diverse tasks and models. We assess five representative reasoning styles—Chain-of-Thought (CoT), Tree-of-Thought (ToT), Algorithm-of-Thought (AoT), Sketch-of-Thought (SoT), and Chain-of-Draft (CoD)—on five reasoning tasks, using 15 open-source models from major families (LLaMA, Qwen, Mistral, Gemma, GPT-OSS, Phi, and DeepSeek) ranging from 270M to 120B parameters. Our large-scale analysis reveals that no single style is universally optimal. We demonstrate that strategy efficacy is highly contingent on both model scale and task type: search-based methods (AoT, ToT) excel in open-ended problems but require large-scale models, while concise styles (SoT, CoD) achieve radical efficiency gains on well-defined tasks. Furthermore, we identify key behavioral patterns: smaller models frequently fail to follow output instructions and default to guessing, while reasoning robustness emerges as a function of scale. Our findings offer a crucial roadmap for selecting optimal reasoning strategies based on specific constraints, We open source the benchmark in `https://anonymous.4open.science/r/StyleBench/`.

## 1 INTRODUCTION

Large Language Models (LLMs) have demonstrated impressive capabilities across a diverse range of tasks, including mathematical reasoning, code generation, and complex question answering (Imani et al., 2023; Wang & Chen, 2023; Tan et al., 2023). A key insight from prior work is that their performance on challenging problems is not merely a function of scale, but is critically dependent on the methods used to guide reasoning (Huang & Yang, 2025). This has spurred the development of sophisticated prompting techniques designed to structure the model's internal reasoning process. Notable among these are *Chain-of-Thought* (CoT) (Wei et al., 2022), which decomposes problems into sequential steps, and more advanced paradigms like *Tree-of-Thought* (ToT) (Yao et al., 2023), which explores multiple reasoning paths in parallel, and *Reasonflux* (Yang et al., 2025b), employing high-level templates to explore potential solutions.

Despite the outstanding capabilities of advanced models like GPT-4o (Agarwal et al., 2025) and DeepSeek (Guo et al., 2025), their application to specialized or highly complex problems often reveals critical limitations. Performance remains highly sensitive to prompt phrasing and frequently necessitates iterative feedback to achieve robust results (Sel et al., 2023). In response, recent work has sought to automate reasoning strategy selection. For instance, Gao et al. (2024) proposed a two-stage meta-reasoning process to identify the optimal style, while Wan et al. (2025) leveraged generated instructions to guide the model. However, these approaches assume the existence of a well-understood mapping between problem types and

optimal reasoning strategies—a foundation that is currently lacking. The existing literature provides an incomplete picture, as evaluations are typically limited to a single reasoning style, a narrow set of tasks, or a small selection of models. This leaves a significant gap in our understanding of how these strategies generalize across different model architectures, problem domains, and computational budgets.

A critical challenge is the trade-off between reasoning depth and efficiency. For simple queries, we expect an LLM to provide a concise, direct answer without a verbose reasoning chain. For complex problems, however, a more elaborate 'thinking' procedure is necessary to achieve high accuracy. This raises the important issue of preventing LLMs from *overthinking* on simple tasks without compromising their ability to reason deeply on hard ones, see e.g., (Chen et al., 2024; Fang et al., 2025; Sui et al., 2025). An ideal reasoning strategy should be both effective and efficient, adapting its cognitive load to the complexity of the problem at hand.

The gaps in generalization understanding and the need for adaptive efficiency motivate the central question of our benchmark:

> *How do contemporary reasoning strategies perform across a diverse suite of tasks, model scales, and architectures, and which approach offers the optimal balance between performance and computational efficiency?*

To address these gaps, we introduce StyleBench, a rigorous and extensive benchmark for evaluating reasoning strategies in LLMs. We systematically assess several representative styles—from simple prompting to complex multi-path searches—across diverse models and tasks. Our work provides clear, empirical guidance and a practical roadmap for selecting the most effective reasoning strategy for a given application.

Our contributions are summarized as follows:

- **Comprehensive Benchmark:** We introduce a large-scale benchmarking framework that systematically evaluates five reasoning styles (Chain of Thought Wei et al., 2022, Tree of Thought Yao et al., 2023, Algorithm of Thought Sel et al., 2023, Sketch of Thought Aytes et al., 2025, Chain of Draft Xu et al., 2025) across five diverse tasks, including mathematical reasoning (Cobbe et al., 2021), question answering (Liu et al., 2020; Talmor et al., 2018), and puzzle-solving.

- **Extensive Model Coverage:** Our evaluation encompasses 15 state-of-the-art open-source LLMs spanning major model families (Qwen Yang et al., 2025a, LlaMA Grattafiori et al., 2024, Mistral Jiang et al., 2024, Gemma Team et al., 2025, GPT-OSS Agarwal et al., 2025, DeepSeek Guo et al., 2025) and scales (270M to 120B parameters).

- **Model–Style Interaction:** We demonstrate that reasoning style efficacy is highly contingent on model architecture and scale, showing that optimal strategy selection is model-dependent.

- **Task–Style Affinities:** We identify strong correlations between task types and effective reasoning strategies. Structured multi-step reasoning (e.g., CoT) excels in mathematical tasks, while branching-based exploration (e.g., ToT, AoT) proves more effective for open-ended puzzles like Game of 24. In-context learning styles (CoD, SoT) perform best on symbolic reasoning and commonsense reasoning tasks.

- **Scaling Laws for Reasoning:** We provide the empirical analysis of how reasoning style performance scales with model size, revealing non-trivial trade-offs between accuracy, latency and efficiency.

## 2 RELATED WORK

### 2.1 REASONING WITH LARGE LANGUAGE MODELS

Recent advances in LLM reasoning have been driven by the development of structured *thinking styles*. Chain-of-Thought (CoT) prompting (Wei et al., 2022) demonstrated that step-by-step reasoning can substan-

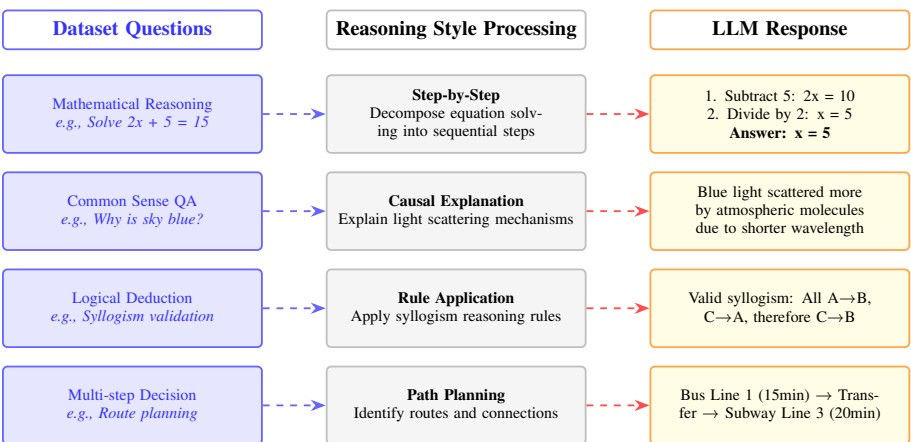

Figure 1: Core Logic and Structural Framework for Reasoning Style Processing

tially improve performance, particularly in mathematical and logical tasks. Building on this idea, Tree-of-Thought (ToT) (Yao et al., 2023) introduced a branching exploration strategy that allows models to consider multiple reasoning paths in parallel. Subsequent paradigms such as Algorithm-of-Thought (AoT) (Sel et al., 2023), Sketch-of-Thought (SoT) (Aytes et al., 2025), and Chain-of-Draft (CoD) (Xu et al., 2025) further extend this direction by incorporating algorithmic priors, lightweight reasoning sketches, or iterative drafting mechanisms into prompts. Other approaches leverage high-level templates or rich contextual information to equip LLMs with more structured reasoning capabilities (Gao et al., 2024; Yasunaga et al., 2023; Yang et al., 2025b). Despite these advances, most existing methods rely on a fixed reasoning style determined in advance, which may be suboptimal across heterogeneous tasks.

## 2.2 BENCHMARKING LLM REASONING

The development of comprehensive benchmarks has been crucial for evaluating the reasoning capabilities of Large Language Models (LLMs). Existing research has largely focused on specialized domains, each requiring distinct reasoning skills. Mathematical reasoning is commonly assessed using benchmarks such as GSM8K (Cobbe et al., 2021), HardMath (Fan et al., 2024) and the more challenging AIME problems. For logical reasoning, datasets like LogiQA (Liu et al., 2020) provide standardized tests, while common-sense reasoning is typically measured by benchmarks such as CommonsenseQA (Talmor et al., 2018). In the domain of code generation, Bigcodebench (Zhuo et al., 2024), HumanEval (Chen et al., 2021) and MBPP (Austin et al., 2021) evaluate functional correctness and algorithmic problem-solving capabilities. More recently, many works have focused on assessing LLMs' performance on puzzle-solving and constrained reasoning tasks. These include problems such as word sorting, Sudoku, and Game of 24, which require structured, multi-step deduction and explore the limits of LLMs' systematic reasoning abilities.

## 3 METHODOLOGY

### 3.1 REASONING STYLES IN LARGE LANGUAGE MODELS

This work evaluates five distinct reasoning methodologies that represent different strategies for structuring the LLM problem-solving process:

**Chain of Thought (CoT)** (Wei et al., 2022) guides models to decompose problems into a sequential series of intermediate steps. By explicitly generating a reasoning trace, this approach significantly improves performance on multi-step tasks like mathematical reasoning.

**Chain of Draft (CoD)** (Xu et al., 2025) emphasizes brevity by constraining models to produce condensed, symbolic reasoning traces. The prompt establishes this format through few-shot examples, leading to responses like '$20 - x = 12, x = 20 - 12 = 8$, ####8' for arithmetic problems.

**Sketch of Thought (SoT)** (Aytes et al., 2025) uses a two-stage process: a trained adapter first identifies the question type, then retrieves relevant few-shot examples to augment the prompt. This encourages concise, symbolic answers while maintaining transparency.

**Tree of Thought (ToT)** (Yao et al., 2023) frames reasoning as a tree search, maintaining multiple parallel reasoning paths (nodes) and pruning less promising branches. This allows for more systematic exploration of the solution space than linear methods.

**Algorithm of Thought (AoT)** (Sel et al., 2023) implements backtracking search, enabling the model to retreat from unproductive paths and explore alternatives, thereby mimicking algorithmic problem-solving.

Example prompts for each style are provided in Table 1, with additional visualizations of each mechanism included in the Appendix.

Table 1: Comparison of Different Thinking Styles for Mathematical Problem Solving

| **Prompt:** | *Solve the equation: 3x + 7 = 22* |
| --- | --- |
| **Thinking Style** | **Processed Prompt** |
| **CoT** | Let me solve this step by step $\cdots$ |
| **CoD** | Think step by step, but only keep a minimum draft for each thinking step, with 5 words at most. Return the answer at the end of the response after a separator ####. |
| **AoT** | Let me explore multiple approaches, and backtrack to the previous one when the current path is incorrect. |
| **SoT** | This is a symbolic reasoning question. I will feed you with some examples to solve this type of question$\cdots$ |
| **ToT** | I'll construct a solution tree: Root: 3x + 7 = 22. Branch through different solution paths and select the optimal approach. |

### 3.2 THE STYLEBENCH BENCHMARK

The StyleBench benchmark was created by writing the question from each dataset in the form of each thinking style, then pass the prompt to the evaluation model. This results in 500 prompts for each model and

each dataset under each thinking style. Examples of prompt entries and model responses can be found in Appendix C. Each thinking style's mechanism is visualized in Appendix A.

### 3.3 BENCHMARK CONSTRUCTION

StyleBench evaluates five distinct reasoning methodologies across a comprehensive suite of 15 open-source language models. The selected models cover a wide range of parameter scales (270M to 120B) and major architectural families—including LLaMA, Qwen, Mistral, Gemma, GPT-OSS, Phi, and DeepSeek—to ensure the broad applicability of our findings. To ensure reproducibility and deterministic outputs, we set the model temperature to 0 for all experiments and collected a single response per model-question pair. Model performance was evaluated by automatically extracting the final answer from each generated response and comparing it against the ground truth.

We categorize the evaluated models into three groups based on scale:

**Small-scale models** (< 5B parameters) include Gemma3-270M (Google), Qwen2.5-0.5B (Alibaba), DeepSeek-R1-Distill-Qwen-1.5B (DeepSeek AI), Gemma-2B (Google), Qwen2.5-3B (Alibaba), and Phi-3-Mini-4K-Instruct-3.8B (Microsoft). This group features models optimized for efficiency, with the distilled DeepSeek model and instruction-tuned Phi-3 providing specific insights into compact reasoning capabilities.

**Medium-scale models** (5B–15B parameters) include Mistral-7B (Mistral), Qwen-7B (Alibaba), Llama3-8B (Meta), and Gemma2-9B (Google). These models strike a practical balance, offering substantial reasoning capabilities while remaining feasible for real-world deployment.

**Large-scale models** (> 15B parameters) include GPT-OSS-20B (OpenAI), Qwen2.5-32B (Alibaba), Llama3-70B (Meta), Qwen2.5-72B (Alibaba), and GPT-OSS-120B (OpenAI). This group enables the study of advanced reasoning emergence at scale, with the recently released GPT-OSS series providing valuable performance baselines for large models.

## 4 RESULTS

We begin by providing a high-level overview of the aggregate performance across different reasoning styles. To enable this comparison, we first compute the mean accuracy of each style over the five benchmark datasets. These scores are then averaged across all models within each of the three size-based groups (small, medium, and large). This analysis highlights several key trends regarding the interaction between model scale, task type, and reasoning strategy. The aggregated results for the three model groups are presented in Figure 2, while detailed accuracy scores for each model, style, and task are reported in Appendix B.

### 4.1 AGGREGATE PERFORMANCE AND SCALING TRENDS

As expected, the performance of all reasoning styles improves with increasing model scale. However, the rate of improvement is not uniform. Search-based strategies like ToT and AoT demonstrate a pronounced scaling law, showing their highest relative advantage on challenging tasks like AIME and Game of 24 *only* when using large-scale models. In contrast, their performance on small and medium models is unremarkable. CoD emerged as the most stable and robust style across all model sizes and tasks.

### 4.2 THE ROLE OF INNATE KNOWLEDGE VS. REASONING

The results on CommonsenseQA highlight the distinction between knowledge retrieval and reasoning. For large models, all reasoning styles perform similarly well, suggesting that the model's inherent knowledge is sufficient to solve the task with even minimal prompting. Conversely, for small and medium models, all

styles struggle profoundly; the best-performing style (SoT for medium models) barely exceeds 6% accuracy. This stands in stark contrast to the large models, where even the worst-performing style (CoT) surpasses 30%, underscoring a vast performance gap driven primarily by model scale.

### 4.3 TASK-STYLE AFFINITIES

Our analysis reveals strong, task-specific affinities for certain reasoning styles, independent of model size.

**GSM8K.** Contrary to our expectation that complex reasoning strategies would prevail, Chain-of-Thought (CoT) consistently outperformed all others across every model group. This indicates that for mathematical problems of this difficulty, a straightforward, stepwise reasoning process is not only sufficient but optimal.

**LogiQA.** SoT proved to be the unequivocally superior strategy, with a significant accuracy margin over all others. We hypothesize that this is because logical reasoning tasks benefit from structured, symbolic reasoning traces. SoT's use of correlated few-shot examples with concise answers allows it to maximize the utility of a constrained context window, efficiently guiding the model to the correct logical conclusion.

## 5 DISCUSSION

To move beyond aggregate metrics and understand the nuanced failures and successes of different reasoning strategies, we conduct a detailed qualitative analysis of model responses. This case study approach allows us to probe the underlying causes of performance difference and address three critical questions: (1) What causes a specific reasoning style to fail on a task where others succeed? (2) To what extent do these failures reflect a fundamental misunderstanding versus a minor, recoverable error? (3) What are the practical implications for selecting an optimal reasoning style for a given problem? Our analysis of these failure modes and success patterns provides crucial insights into the inner workings of LLM reasoning.

### 5.1 KEY FINDINGS

> **Findings:** On high-difficulty tasks, small models typically do not exhaust the available token budget. Instead, they tend to produce an answer regardless of correctness. This behavior suggests that when an LLM lacks the capability to solve a problem, it often defaults to *guessing* rather than abstaining.

We analyze the average token consumption of each reasoning style across different models on two challenging tasks, AIME and Game24, as visualized in Figure 3.

Contrary to what might be expected, Figure 3 shows that smaller models (with the exception of Deepseek-Distill) do not consistently consume more tokens than large models (e.g., Qwen-72B or GPT-OSS-120B) in these tasks. This is particularly notable in high-difficulty tasks like Game24, which requires models to exhaustively search for combinations of elementary arithmetic operators using four given numbers to reach the target of 24. Although smaller models struggle to find correct solutions (as shown in Figure 2), they frequently complete their generation without hitting the predefined token budget limit.

To investigate this further, we present examples of responses from Qwen-3B and Llama-3-8B on the AIME and Game24 datasets, respectively, in Appendix D.1. Both responses are incorrect, yet each model produced a final answer and terminated generation naturally, rather than being truncated for exceeding limits.

In the Qwen-3B response for AIME (see the first example), the model's reasoning is correct until Equation 2. However, it makes a critical error during verification: after setting $n = 60$ and obtaining $x = 156$, it incorrectly accepts this result despite the constraint that $x \leq n$. This illustrates that while small models can sometimes generate high-level rationales, their inability to avoid subtle but decisive logical flaws

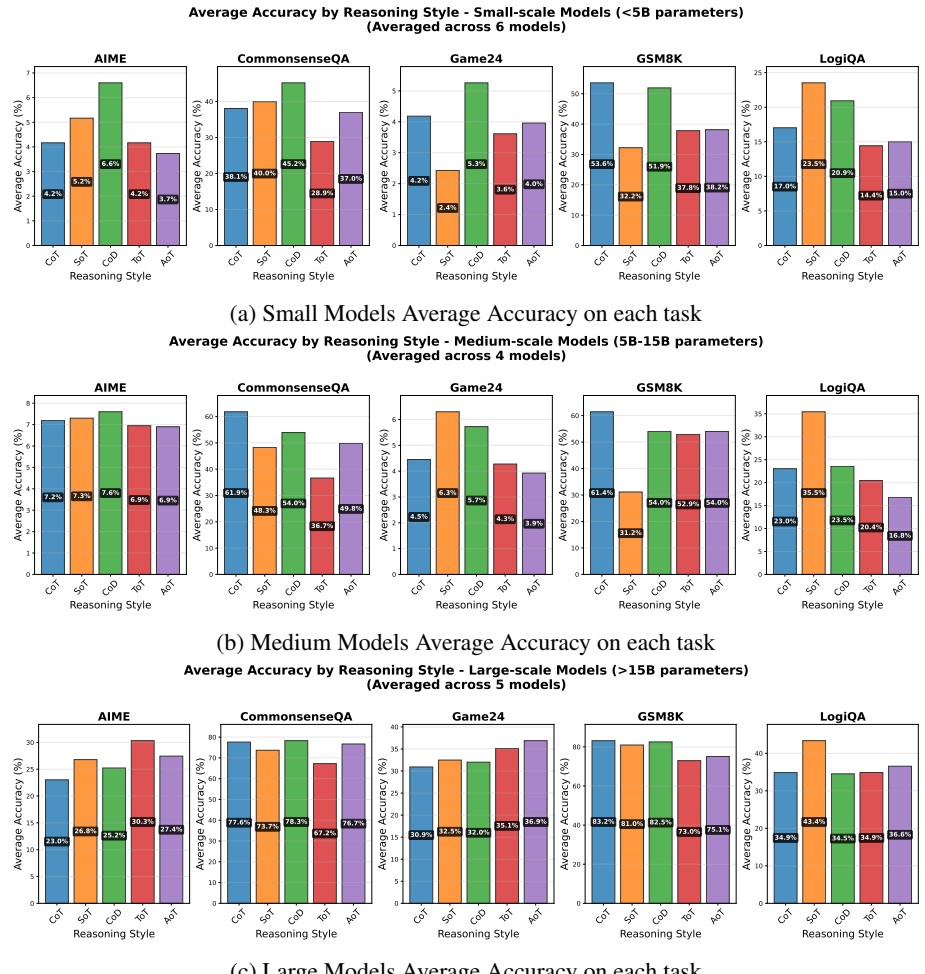

(a) Small Models Average Accuracy on each task

(b) Medium Models Average Accuracy on each task

(c) Large Models Average Accuracy on each task

Figure 2: Overall Accuracy rate for each group of models across five tasks

ultimately prevents them from arriving at the correct solution. The second response, from Llama-3-8B on Game24, reveals a different failure mode. The model terminates after outputting the expression $12 + 12$, which violates the core rules of the task, as it uses only two of the four provided numbers. Furthermore, its intermediate reasoning steps (Step 2 and Step 3) are entirely disconnected from the final output. This suggests that for smaller models, advanced prompting instructions do not reliably lead to coherent, multi-step reasoning; instead, the models often produce superficially structured but ultimately nonsensical rationales before guessing an answer.

These patterns indicate that the primary bottleneck for small and medium-sized LLMs on complex tasks is not *a lack of generative capacity* (as they do not exhaust token budget) but a fundamental deficiency in *reasoning capability*. This phenomenon aligns with recent research on LLM behavior, such as Fu et al. (2025) showing that LLMs can exhibit unjustified confidence in incorrect answers during reasoning processes.

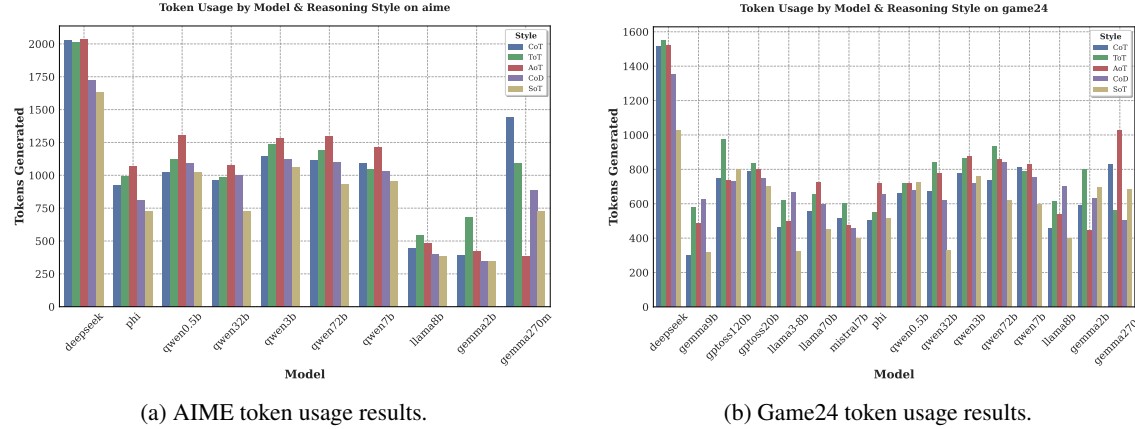

(a) AIME token usage results.        (b) Game24 token usage results.

Figure 3: Token usage comparison of different models on two datasets.

> **Findings:** Following output format instructions is a skill that improves with model size. Small models often ignore formatting, making evaluation harder.

Our case study (see D.2) reveals that this formatting issue is widespread. Even with clear instructions to use the `\boxed{answer}` format, smaller models frequently ignore this directive. This inconsistency creates major challenges for automated evaluation, requiring additional processing to extract answers correctly. The problem extends beyond mathematical tasks. On LogiQA and CommonsenseQA, we often find smaller models adding unexpected characters around answers—like '$< A >$' or '$< 2 >$'—instead of following the specified format. This behavior appears to reflect patterns learned during pretraining that smaller models lack the capacity to override when given explicit instructions.

> **Findings:** Search-based methods like AoT and ToT work well for open problems like Game24 by trying many solutions, but incur higher token usage. For clearer tasks like LogiQA, concise methods like SoT and CoD give short, accurate answers fast.

The responses show clear strengths and weaknesses for each reasoning style. In the Game24 problem (D.3), AoT found a solution by flexibly trying different math operations. In contrast, ToT failed because its first guess was wrong and it could not recover without using many more tokens. This shows a key problem with tree-search: it needs good first guesses to work well. On the same Game24 problem, CoT, CoD, and SoT all failed. They tried random number combinations instead of searching in a smart way. This shows that methods without a built-in search strategy often guess poorly on open-ended puzzles.

However, CoD and SoT work very well for structured tasks like LogiQA (D.4). They produced much shorter answers than CoT (16% and 94% shorter) because they work in different ways:

- CoD uses knowledge from its training to give more direct answers.
- SoT skips unnecessary steps by connecting ideas quickly.

Both methods kept high accuracy while being much more efficient. This makes them very useful for real-world applications where speed and cost matter.

> **Findings:** Effective reasoning on challenging problems is strongly dependent on model scale. Large models can succeed with a variety of strategies, while smaller models require specialized, advanced prompting to achieve comparable performance.

This scaling effect is evident in our case studies (Appendix E.1 and E.2), which evaluate reasoning styles across the Qwen series (3B to 72B) and GPT-OSS-120B on the Game24 task. The smallest models (Qwen-3B/7B) failed completely. Qwen-32B generated a solution, but it violated the game rules. Only the larger models succeeded: Qwen-72B produced two valid solutions (via CoT and AoT), and GPT-OSS-120B produced one correct solution (via SoT). Critically, large models consistently generated meaningful solution attempts, whereas smaller models often failed to make progress or became stuck in unproductive loops, highlighting a fundamental capability gap.

## 5.2 KEY QUESTIONS

**Can LLMs autonomously select the most effective reasoning style for a given problem?**

Our findings indicate that this meta-reasoning capability remains emergent and is not readily achievable through standard supervised fine-tuning (SFT). We fine-tuned a Qwen-7B model to identify optimal reasoning styles, providing explicit rationales for each selection. The model failed to develop a robust selection strategy, instead defaulting to shallow memorization of the training distribution. This resulted in a pathological bias toward consistently selecting Chain-of-Draft (CoD) across diverse problems, yielding no substantial advantage over using any single fixed style. This behavior aligns with known limitations of SFT, particularly model hallucination and shallow pattern matching Ren & Sutherland (2024). Crucially, these results reveal that current fine-tuning approaches enable only a superficial association with style selection rather than a genuine, contextual understanding of which strategy best fits a given problem. Experimental details are provided in Appendix G.

## 6 CONCLUSION

In this work, we introduced StyleBench, a comprehensive benchmark for systematically evaluating five reasoning styles (CoT, ToT, AoT, SoT, CoD) across five diverse tasks and 15 language models spanning 270M to 120B parameters. Our large-scale analysis yields several crucial insights that advance our understanding of reasoning in LLMs. Our findings demonstrate that reasoning strategy effectiveness is highly contingent on both task requirements and model capabilities, with no single approach dominating across all scenarios. We identified distinct behavioral patterns across model scales: smaller models frequently disregard formatting instructions and default to guessing, while larger models exhibit more reliable instruction-following and systematic reasoning capabilities. Most significantly, we observed a substantial performance gap that correlates strongly with model scale, confirming that fundamental reasoning abilities—including the capacity for meta-reasoning about strategy selection—emerge primarily with increased model size.

These results provide a practical framework for optimal strategy selection: search-based methods (ToT, AoT) excel for complex, open-ended problems (e.g., Game24) with capable models, while concise approaches (SoT, CoD) offer superior efficiency for well-structured tasks (e.g., LogiQA, CommonsenseQA) or resource-constrained environments. However, our experiments also reveal limitations: attempts to teach style selection via supervised fine-tuning resulted in shallow memorization rather than genuine strategic understanding, highlighting the need for more sophisticated approaches to meta-reasoning.

By establishing these scaling laws, task-style affinities, and the current boundaries of adaptive reasoning, StyleBench provides both a valuable evaluation framework and a strategic foundation for developing more efficient, robust, and self-aware reasoning systems in language models.

ETHICS STATEMENT

We adhere to the ICLR 2026 Code of Ethics.

**LLM Usage Disclosure:** We used a large language model only for writing assistance (specifically, polishing grammar, improving clarity, and refining wording). We did not use LLMs for conceiving research ideas, designing experiments, or drawing conclusions. All experimental design, reasoning styles, task definitions, results, and analyses are our own work.

**Data / Tasks Used:** Our evaluations are conducted on publicly available, open-source benchmark datasets (e.g., GSM8K, CommonsenseQA, LogiQA, AIME, Game24 from HuggingFace) and models (e.g., Qwen series and Llama series LMs). We did not collect new human subject data. All dataset usage complies with their respective licenses.

**Reproducibility & Anonymous Link:** To facilitate reproducibility, we provide an anonymous link to the benchmark evaluation data and processed dataset splits; the link is hosted without any identifying metadata and does not contain author affiliations or names.

**Fairness, Bias, and Misuse Risks:** We recognize that LLMs may produce incorrect or biased reasoning, or perform differently across reasoning styles and tasks. In our work, we systematically evaluate across multiple reasoning styles and diverse datasets to identify both strengths and failure modes.

**Privacy / Confidentiality:** No private or sensitive personal data is used or generated. All data and outputs are from benchmark tasks; we ensure that none of the datasets or evaluation artifacts reveal identifying information. All processing is done under secure protocols.

**Responsibility & Integrity:** We, the authors, take full responsibility for all content, including content polished by LLM. Any errors or misrepresentations introduced during text polishing have been checked and corrected.

REPRODUCIBILITY STATEMENT

We have made significant efforts to ensure reproducibility of our experiments. All datasets used are publicly available open-source benchmarks, and we provide full descriptions of the reasoning style definitions, data processing steps, and experimental protocols in the main manuscript and in the appendix.

We also supply an anonymous link to our benchmark's processed data and README file, hosted at:

https://anonymous.4open.science/r/StyleBench/README.md

This link contains no identifying metadata or author names/affiliations.

Implementation details, hyperparameters, and any scripts needed to reproduce our main results are referenced in the README file and supplementary materials.

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

# A    THINKING STYLES

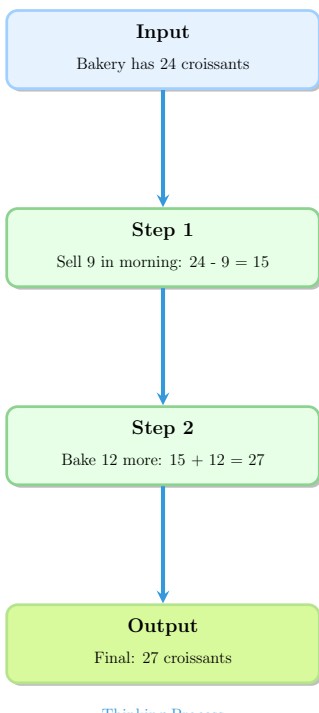

(a) Chain-of-Thought (CoT): Linear step-by-step reasoning

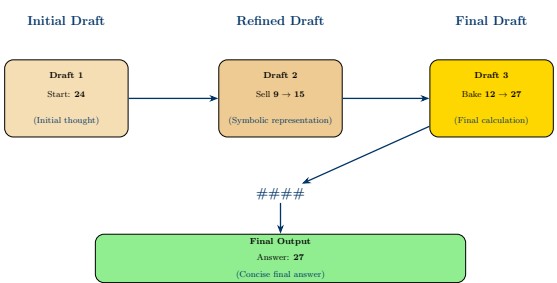

(b) Chain-of-Draft (CoD): Iterative refinement process

Figure 4: Sequential reasoning methodologies: CoT follows a linear progression while CoD employs iterative refinement of drafts.

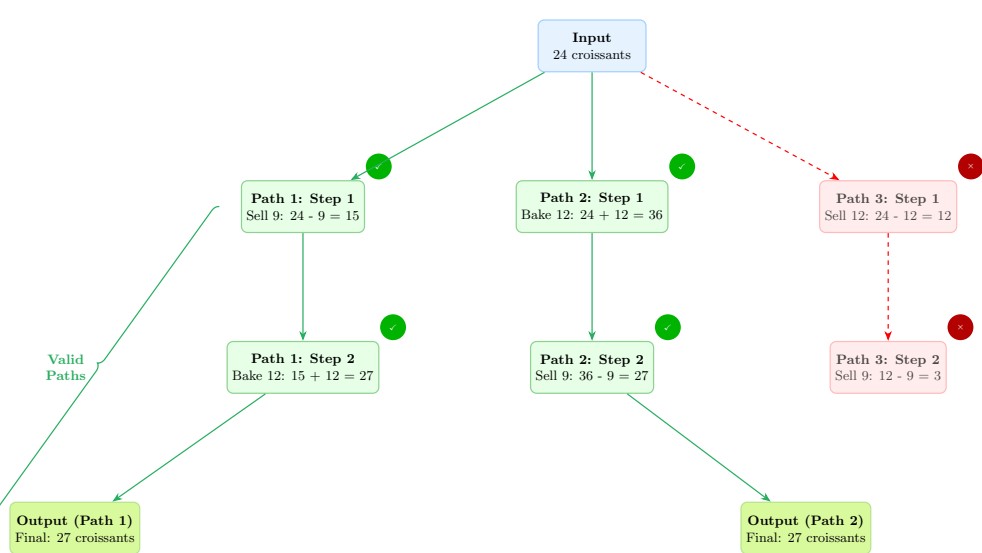

(a) Algorithm-of-Thought (AoT): Backtracking exploration

(b) Tree-of-Thought (ToT): Branching and pruning

Figure 5: Exploratory reasoning methodologies: ToT explores multiple paths with selective pruning, while AoT employs systematic backtracking to explore alternative approaches.

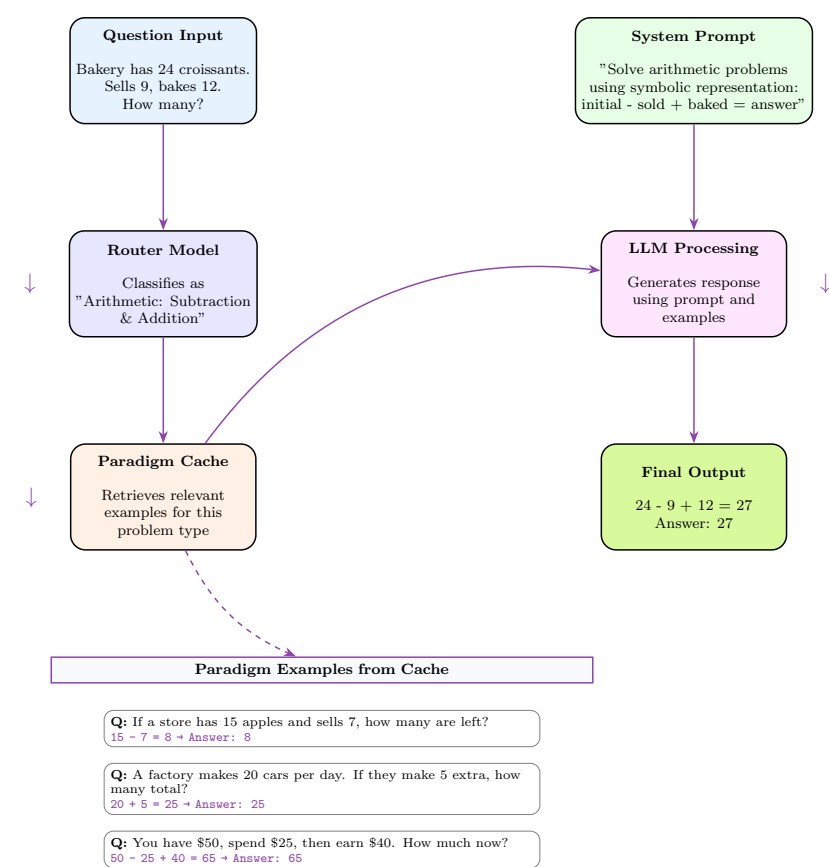

Figure 6: Sketch-of-Thought (SoT): Router-based paradigm selection with exemplar retrieval. The method classifies the input problem, retrieves relevant examples from a paradigm cache, applies targeted prompts, and generates responses through structured LLM processing.

# B  OVERALL ACCURACY SCORE

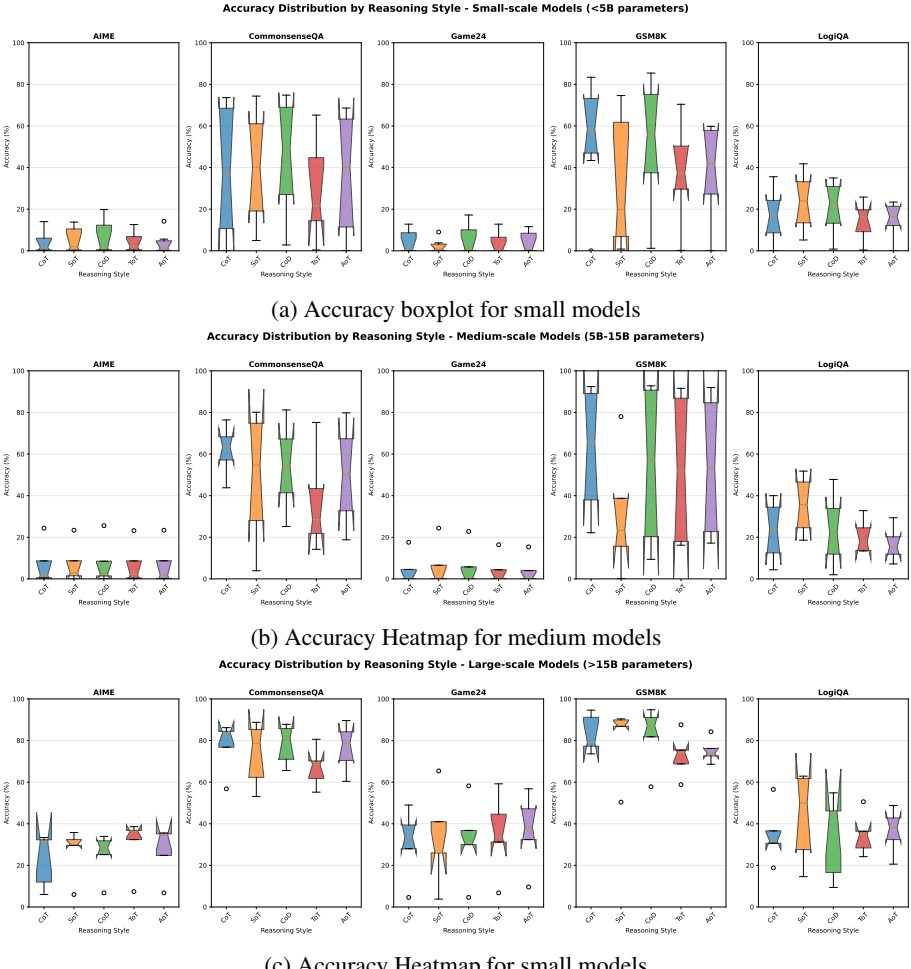

(a) Accuracy boxplot for small models

(b) Accuracy Heatmap for medium models

(c) Accuracy Heatmap for small models

Figure 7: Boxplot for three groups of model accuracy.

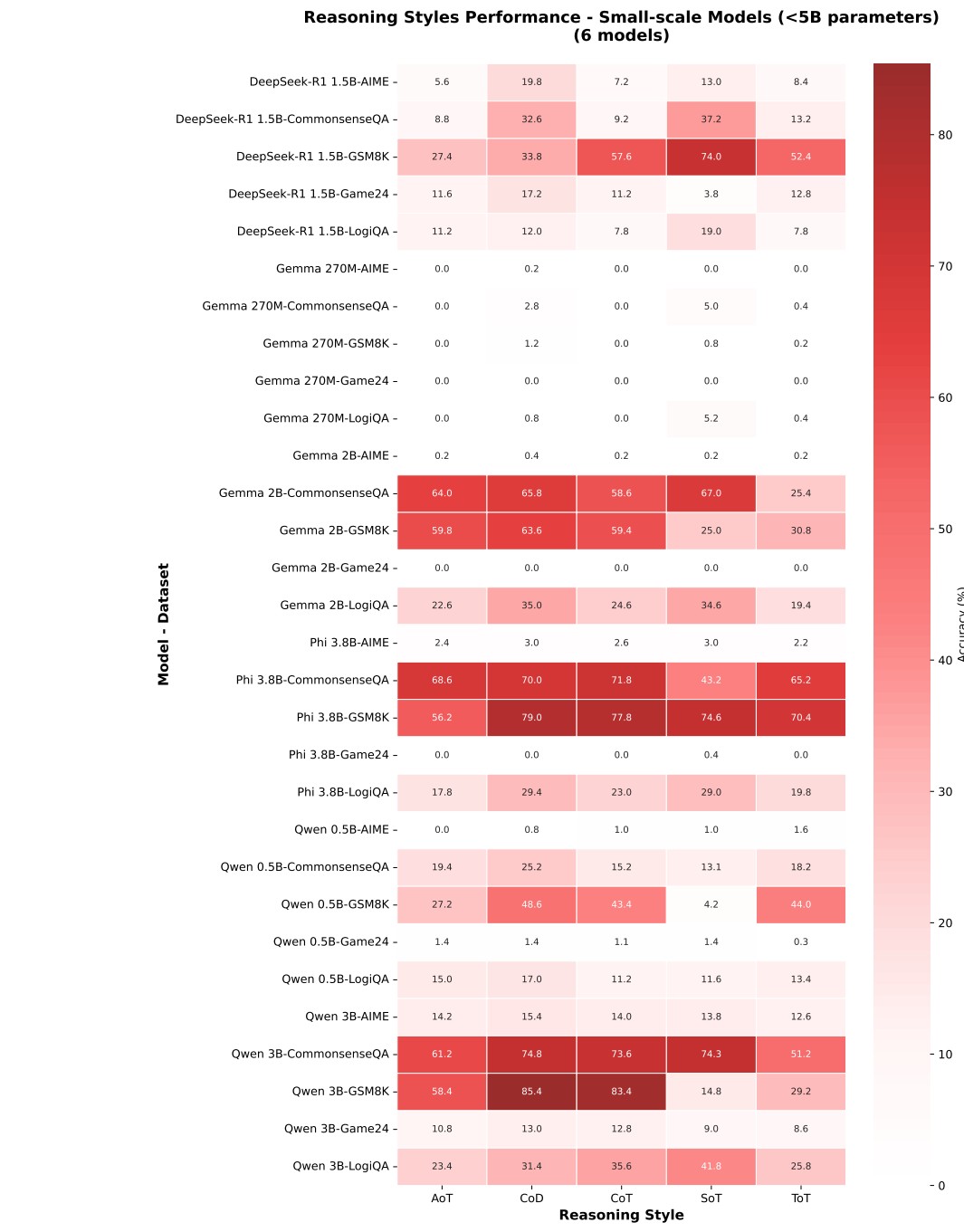

Figure 8: Accuracy Heatmap for small models

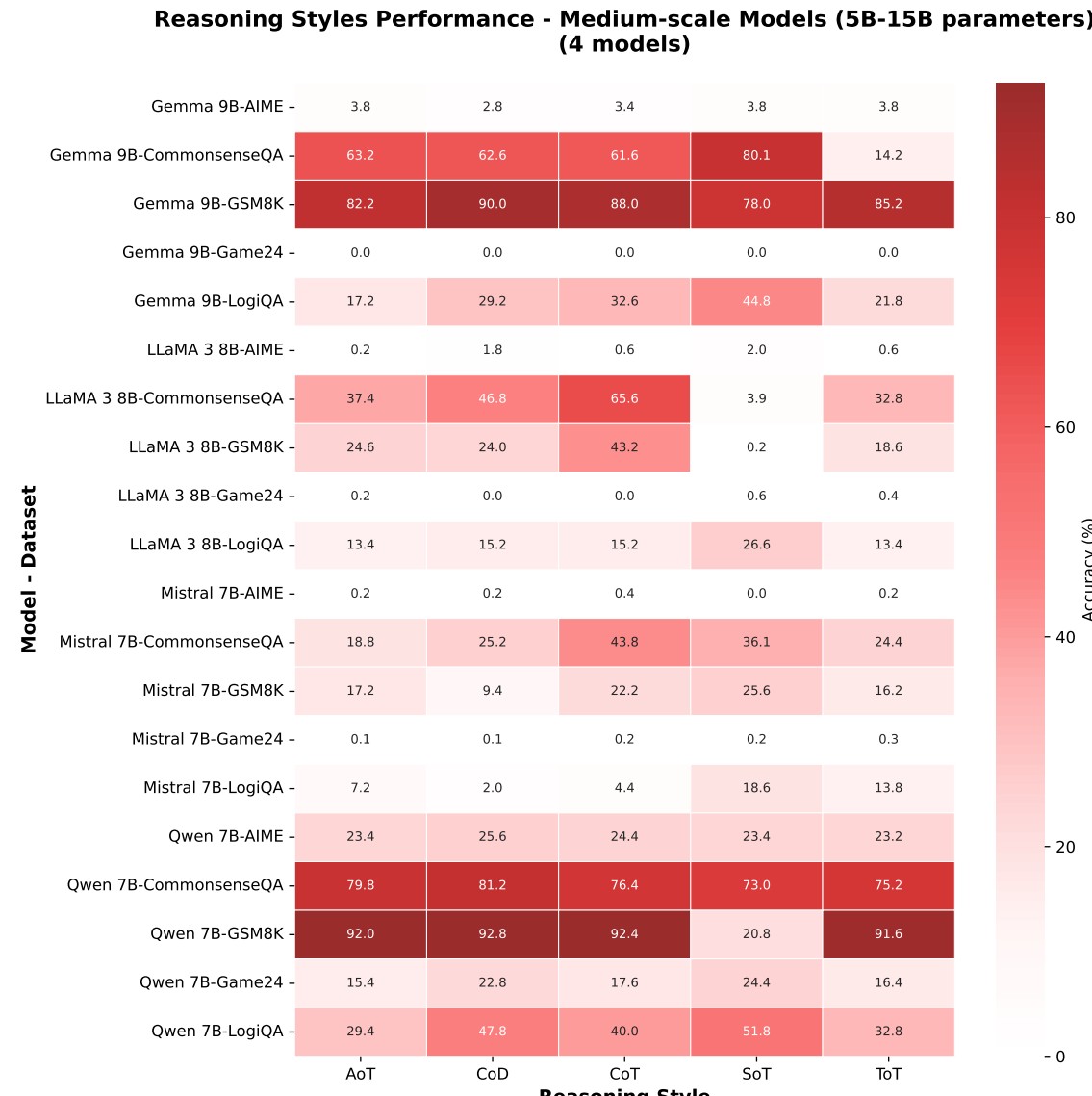

Figure 9: Accuracy Heatmap for medium models

Figure 10: Accuracy Heatmap for large models

## C  SAMPLE PROMPTS BY REASONING STYLE

Here we provide the sample prompts for each thinking style and we choose the CommonsenseQA dataset for example.

---

**Sample CommonsenseQA Problem**

Bob the lizard lives in a warm place with lots of water. Where does he probably live? Options: A) rock B) tropical rainforest C) jazz club D) new mexico E) rocky places

---

**Chain of Thought (CoT)**

**System:** You are a careful math solver. Think step by step and show all your work clearly. Explain your reasoning process, then provide your final answer. At the end, put your final numerical answer in \boxed{< answer >} format.
**User:** Problem: Bob the lizard lives in a warm place with lots of water. Where does he probably live? Options: A) rock B) tropical rainforest C) jazz club D) new mexico E) rocky places
Let's work through this step by step, showing all calculations.
Please solve this step by step, showing all your work and reasoning. Put your final numerical answer in \boxed{< think >} at the very end.

---

**Tree of Thoughts (ToT)**

**System:** Imagine three different experts are solving this math problem together. Each expert contributes their reasoning step by step. They discuss, check each other's work, and collaborate. Show this collaborative problem-solving process with clear reasoning. At the end, put the final agreed-upon answer in \boxed{< think >} format.
**User:** Problem: Bob the lizard lives in a warm place with lots of water. Where does he probably live? Options: A) rock B) tropical rainforest C) jazz club D) new mexico E) rocky places
Show how the three experts would collaborate on this problem, with each contributing their reasoning and checking each other's work.
Please solve this step by step, showing all your work and reasoning. Put your final numerical answer in \boxed{< think >} at the very end.

---

**Algorithm of Thoughts (AoT)**

**System:** Use two-phase reasoning to solve this problem: Phase 1 - Forward Analysis: Think step by step through the problem from start to finish. Phase 2 - Verification: Work backward from your answer to double-check it's correct. Show both phases of your reasoning clearly. At the end, put your final answer in \boxed{< think >} format.
**User:** Problem: Bob the lizard lives in a warm place with lots of water. Where does he probably live? Options: A) rock B) tropical rainforest C) jazz club D) new mexico E) rocky places
Use both forward analysis and backward verification. Show both phases of your reasoning clearly.
Please solve this step by step, showing all your work and reasoning. Put your final numerical answer in \boxed{< think >} at the very end.

## Chain of Draft (CoD)

**System:** Use a Draft→Refine workflow to solve problems step by step. Show your draft solution, then refine and improve it with detailed reasoning. At the end, put your final answer in \boxed{< think >} format.
**User:** 2015 is coming in 36 hours. What is the date one week from today in MM/DD/YYYY?
**Assistant:**{< think >} 36h = 1.5 days; today: 2015/01/01 - 1.5 = 12/30/2014; 12/30/2014 + 7 = 01/06/2015. {< /think >} \boxed{01/06/2015}
...
[22 more similar examples with {< think >} format responses]
...
**User:** Bob the lizard lives in a warm place with lots of water. Where does he probably live? Options: A) rock B) tropical rainforest C) jazz club D) new mexico E) rocky places
Please solve this step by step, showing all your work and reasoning. Put your final numerical answer in \boxed{< think >} at the very end.

## Symbolic of Thoughts (SoT)

**System:**
**Role & Objective**
You are a reasoning expert specializing in **Chunked Symbolism**, a cognitive reasoning technique that organizes numerical reasoning into structured steps. Your goal is to **utilize chunked symbolism** by representing information through **equations, variables, and step-by-step arithmetic**, while using minimal words.
Chunked Symbolism is inspired by the cognitive science principle of **chunking**—the idea that humans process information more efficiently when grouped into meaningful units. Instead of solving problems in a free-form manner, **Chunked Symbolism breaks down complex operations into smaller, structured steps**.
This method is particularly effective for:
- **Mathematical problems** (arithmetic, algebra, physics, engineering)
- **Symbolic reasoning** (logic-based computations, formula derivations)
- **Technical calculations** (financial modeling, physics simulations, unit conversions)
**How to Apply Chunked Symbolism**
**Step-by-Step Guide**
1. **Identify Variables** – Extract relevant numerical values and define variables.
2. **Write Equations** – Represent the solution using **explicit mathematical formulas**.
3. **Perform Step-by-Step Computations** – Solve in **small, logical steps**, keeping each line clear.
4. **Label Units** – Maintain **consistent unit representation** to prevent ambiguity.
5. **Final Answer Formatting** – Present the answer in the **provided format** for clarity.
**Rules & Directives**
1. **Use Equations & Variables**
- Define variables before computation.
- Always use **explicit equations** to represent reasoning.
2. **Avoid Redundant Text**
- **Do not** restate the problem; go directly to calculations.
- Use **minimal context** only if it aids understanding.
3. **Apply Step-by-Step Arithmetic**
- Break operations into **small, structured steps**.

- Ensure each line contains only **one computation** for clarity.

4. **Output Format**

- Use the exact structured format:

```

< think > [shorthand reasoning] < /think > \boxed{[Final answer]} ``` - The **final answer must be boxed**. - **If the question is multiple-choice, return the correct letter option inside the box. - Use minimal words in your response.**

**User:** A car accelerates at 2.5 m/s$^2$ for 10 seconds. If its initial velocity was 15 m/s, what is its final velocity?

**Assistant:** < think > a = 2.5 m/s$^2$ t = 10 s vi = 15 m/s vf = 15 + (2.5 × 10) vf = 40 m/s < /think > \boxed{40}

**User:** If a product costs \$120 and there is a 15% discount, what is the final price? Choices: A) \$10 B) \$97 C) 102

**Assistant:**< think > op = 120 d = 15% dp = 120 × (15 / 100) = 18 fp = 120 - 18 = 102 < /think > \boxed{C}

**User:** Question: A circuit has a voltage of 12V and a resistance of 4Ω. What is the current?

**Assistant:** < think > V = 12V R = 4Ω I = 12 / 4 = 3A < /think > \boxed{3}

**User:** Bob the lizard lives in a warm place with lots of water. Where does he probably live? Options: A) rock B) tropical rainforest C) jazz club D) new mexico E) rocky places

Please solve this step by step, showing all your work and reasoning. Put your final numerical answer in \boxed{< think >} at the very end.

# D  SAMPLE RESPONSE

## D.1  INCORRECT RESPONSE EXAMPLES

This is a response produced by Qwen2-3B model.

---

**LLM Response**

**Question:** The pages of a book are numbered 1 through $n$. When the page numbers of the book were added, one of the page numbers was mistakenly added twice, resulting in an incorrect sum of 1986. What was the number of the page that was added twice?

**Ground Truth:** 33

**LLM Response:**

Here's how to solve the problem step-by-step:

**1. Formula for the sum of an arithmetic series:**

The sum of the first $n$ natural numbers is given by the formula:

$$S = \frac{n(n+1)}{2} \tag{1}$$

where $S$ is the sum of the series.

**2. Setting up the equation:**

- Let 'n' be the total number of pages in the book.
- The sum of the page numbers, without the duplicate, would be $\frac{n(n+1)}{2}$.
- Since one page number was added twice, the incorrect sum is: $\frac{n(n+1)}{2} + x$

where 'x' is the page number that was added twice.

We know this incorrect sum is 1986:

$$\frac{n(n+1)}{2} + x = 1986 \tag{2}$$

**3. Finding a pattern:**

We need to find a value for 'n' and 'x' that satisfies this equation. Notice that 1986 is close to a perfect square. Let's try some values of 'n' and see if we can find a solution.

**4. Trial and Error:**

- If $n = 60$, then $\frac{n(n+1)}{2} = \frac{60(61)}{2} = 1830$.
- This means $x = 1986 - 1830 = 156$.

**5. Verification:**

Let's check if our answer works:

---

- Is 156 a page number in the book? Yes, it could be.

Therefore, the page number that was added twice is 156.

$$\boxed{156}$$

This is a response produced by Llama3-8B model on Game24 task.

### LLM Response: Game of 24

**Puzzle:** Use the numbers 3, 3, 12, 12 exactly once each with operations $+, -, \times, \div$ and parentheses to make 24.
**Ground Truth:** Multiple valid solutions exist (e.g., $(12 \div 3) \times (3 + 3) = 4 \times 6 = 24$)
**LLM Response:**
Here's how we can solve this puzzle:
**1. Look for obvious pairs:**

- Notice that $12 \times 12 = 144$, which is too big.
- Let's try $3 \times 12 = 36$, which is also too big.

**2. Consider division:**

- Since we need to get to 24, let's see if we can divide one of the larger numbers by the smaller ones: $12 \div 3 = 4$.

**3. Build from there:**

- Now we have 4, 3, 3, and 12. We can use the 4 and 3 to get 12: $4 \times 3 = 12$.

**4. Combine the results:**

- We have $12 + 12 = 24$.

**Therefore, the solution is:**
$$\boxed{12 + 12}$$

### D.2 INCORRECT FORMAT RESPONSE EXAMPLE

### LLM Response

**Problem:** Let $m$ be the smallest integer whose cube root is of the form $n + r$, where $n$ is a positive integer and $r$ is a positive real number less than $\frac{1}{1000}$. Find $n$.

**Ground Truth:** $n = 19$

**LLM Response (Chain of Thought):**

Let's start by writing the equation:

$$m^{1/3} = n + r$$

where $n$ is a positive integer and $r$ is a positive real number less than $\frac{1}{1000}$.

Since $r$ is less than $\frac{1}{1000}$, we can write:

$0 < r < \frac{1}{1000}$

We can also write:

$n + r = m^{1/3}$

Subtracting $n$ from both sides gives:

$r = m^{1/3} - n$

Since $r$ is a positive real number less than $\frac{1}{1000}$, we can see that:

$0 < m^{1/3} - n < \frac{1}{1000}$

Adding $n$ to both sides gives:

$n < m^{1/3} < n + \frac{1}{1000}$

Since $n$ is a positive integer, we can see that:

$n < m^{1/3} < n + 1$

This means that $m^{1/3}$ is an integer that is one more than a multiple of $n$. The smallest such integer is $n + 1$, so:

$m^{1/3} = n + 1$

Cubing both sides gives:

$m = (n + 1)^3$

Expanding the right-hand side gives:

$m = n^3 + 3n^2 + 3n + 1$

We want to find the smallest integer $m$ that satisfies this equation. We can start by trying small values of $n$:

- If $n = 1$, then $m = 1^3 + 3(1)^2 + 3(1) + 1 = 8$
- If $n = 2$, then $m = 2^3 + 3(2)^2 + 3(2) + 1 = 27$
- If $n = 3$, then $m = 3^3 + 3(3)^2 + 3(3) + 1 = 56$
- If $n = 4$, then $m = 4^3 + 3(4)^2 + 3(4) + 1 = 85$

We can see that $m = 27$ is the smallest integer that satisfies the equation. Therefore, $n = 2$.

Final Answer: The final answer is 2. I hope it is correct.

## D.3 COMPARISON BETWEEN DIFFERENT STYLES

---

**LLM Responses: Game of 24 - Multiple Reasoning Styles**

**Problem:** Use the numbers 2, 3, 13, 13 exactly once each with operations $+, -, \times, \div$ and parentheses to make 24.

**Ground Truth:** Valid solution exists: $(13 \times 3 - 13) - 2 = 24$

**Chain of Thought (CoT) Response:**
Let's solve the puzzle step by step using the numbers 2, 3, 13, and 13. [The response shows 8 different attempts, all yielding incorrect results, then concludes:]
After trying various combinations, it seems that none of the expressions we have formed using the numbers 2, 3, 13, and 13 with the allowed operations and parentheses result in 24.
**Conclusion:** Since we have exhausted all reasonable combinations and none of them yield 24, we conclude that there is no solution for this set of numbers.
$\boxed{\text{No solution}}$

**Tree of Thoughts (ToT) Response:**
[Shows collaborative discussion between three experts, trying various combinations like $(13 + 13) \times 2 - 3 = 49$, $13 \times (13/2 - 3) = 45.5$, $(13 + 13) - (2 \times 3) = 20$, etc.]
**Final Answer:** After trying multiple combinations, it seems that we cannot form an expression that evaluates to 24 using the numbers 2, 3, 13, and 13 with the allowed operations. $\boxed{\text{No solution}}$

**Analysis of Thoughts (AoT) Response [CORRECT]:**
**Phase 1 - Forward Analysis:**
We are given the numbers 2, 3, 13, and 13. Let's consider different combinations:
**1. Multiplication and Addition:**

- $13 \times 2 = 26$
- $26 - 3 = 23$ (not 24)
- $26 - 13 = 13$ (not 24)

**2. Multiplication and Subtraction:**

- $13 \times 3 = 39$
- $39 - 13 = 26$
- $26 - 2 = 24$ (this looks promising)

**Phase 2 - Verification:**
Let's verify the expression $(13 \times 3 - 13) - 2$:

$$13 \times 3 = 39$$
$$39 - 13 = 26$$
$$26 - 2 = 24 \quad \checkmark$$

$\boxed{(13 \times 3 - 13) - 2}$

**Chain of Draft (CoD) Response:**
[Shows 8 draft attempts including $(13+13-2\times3) = 20$, $(13-3)\times(2+13) = 150$, $(13+13)\div2+3 = 16$, etc.]

After trying multiple combinations, it seems that it is not possible to form an expression using the numbers 2, 3, 13, 13 that evaluates to 24.

No solution

**Structured of Thoughts (SoT) Response:**
$a = 2, b = 3, c = 13, d = 13$
Step 1: Consider possible combinations of operations.
Step 2: Test $(a + b) \times (c - d)$ $(2 + 3) \times (13 - 13) = 5 \times 0 = 0$ (Not 24)
Step 3: Test $(a \times b) + (c - d)$ $(2 \times 3) + (13 - 13) = 6 + 0 = 6$ (Not 24)
Step 4: Test $(a \times b) + c - d$ $(2 \times 3) + 13 - 13 = 6 + 13 - 13 = 6$ (Not 24)
Step 5: Test $(a + b) \times c \div d$ $(2 + 3) \times 13 \div 13 = 5 \times 13 \div 13 = 5$ (Not 24)
Step 6: Test $(a + b) \times (c \div d)$ $(2 + 3) \times (13 \div 13) = 5 \times 1 = 5$ (Not 24)
Step 7: Test $(a \times c) + (b \times d)$ $(2 \times 13) + (3 \times 13) = 26 + 39 = 65$ (Not 24)
Step 8: Test $(a \times c) - (b \times d)$ $(2 \times 13) - (3 \times 13) = 26 - 39 = -13$ (Not 24)
Step 9: Test $(a \times d) + (b \times c)$ $(2 \times 13) + (3 \times 13) = 26 + 39 = 65$ (Not 24)
Step 10: Test $(a \times d) - (b \times c)$ $(2 \times 13) - (3 \times 13) = 26 - 39 = -13$ (Not 24)
[... continues with systematic enumeration of 30+ more combinations ...]
Step 40: Test $(a \div c) - (b \times d)$ $(2 \div 13) - (3 \times 13) = 0.1538 - 39 = -38.8462$ (Not 24)
**Result:** Response was truncated due to length limits. No final answer was provided in the original response.

## D.4 REASONING STYLE COMPARISON: CONCISENESS

**LLM Responses: Commonsense QA - Reasoning Style Comparison**

**Problem:** Reading newspaper one of many ways to practice your what?
**Options:** A) literacy    B) knowing how to read    C) money    D) buying    E) money bank
**Ground Truth:** A) literacy

**Chain of Thought (CoT) Response:** *[Length: 1,192 characters]*
This problem is not a numerical or mathematical problem but rather a question about understanding the context and meaning of the options provided. Let's analyze the question and the options step by step:
1. The question asks, "Reading newspaper one of many ways to practice your what?" 2. We need to determine what reading a newspaper helps to practice.
Now, let's analyze each option:
A) literacy: Literacy refers to the ability to read and write. Reading a newspaper is indeed a way to practice and improve one's literacy skills.
B) knowing how to read: This is similar to literacy but more specific to the act of reading. While reading a newspaper does involve reading, the term "literacy" encompasses a broader set of skills including reading comprehension and writing.
C) money: Reading a newspaper does not directly relate to practicing money management or understanding.
D) buying: Reading a newspaper does not directly relate to practicing the act of buying.
E) money bank: This option is unrelated to reading a newspaper.
Given the analysis, the most appropriate answer is A) literacy, as reading a newspaper is a way to practice and improve one's literacy skills.

$\boxed{A}$

**Tree of Thoughts (ToT) Response:** *[Length: 1,912 characters]*
**Expert 1:** Let's start by analyzing the question. The question asks, "Reading newspaper one of many ways to practice your what?" We need to identify what reading a newspaper helps to practice. Reading a newspaper involves understanding written text, which is a key component of literacy.
**Expert 2:** I agree with Expert 1. Let's look at the options provided: [analyzes all options] Option A, "literacy," directly aligns with the idea of reading and understanding written text.
**Expert 3:** I concur with both of you. Let's eliminate the options that are clearly unrelated: [eliminates options C, D, E] This leaves us with options A and B. While "knowing how to read" (Option B) is a part of literacy, the broader term that encompasses both reading and writing is "literacy" (Option A).
$\boxed{A}$

**Analysis of Thoughts (AoT) Response:** *[Length: 1,779 characters]*
**Phase 1 - Forward Analysis:**
1. **Understanding the Problem:** The problem asks about what reading a newspaper helps to practice.
2. **Analyzing Each Option:** [Systematically analyzes options A through E]
3. **Selecting the Best Option:** Both A and B are valid, but "literacy" (option A) is a broader term that encompasses the ability to read and write, which is what reading a newspaper helps to practice.
**Phase 2 - Verification:** [Works backward from selected option to verify correctness]
$\boxed{A}$

**Chain of Draft (CoD) Response:** *[Length: 1,005 characters - 16% shorter]*
To solve this problem, we need to determine which option best fits the context of the question: "Reading newspaper one of many ways to practice your what?"
Let's analyze each option:
A) literacy: Literacy refers to the ability to read and write. Reading a newspaper is indeed a way to practice and improve one's literacy skills.
B) knowing how to read: This option is somewhat similar to literacy, but it is more specific to the act of reading. While reading a newspaper does involve knowing how to read, the term "literacy" encompasses a broader set of skills, including reading comprehension and writing.
C) money: Reading a newspaper has no direct relation to practicing money management or understanding.
D) buying: Reading a newspaper does not involve practicing the act of buying.
E) money bank: This option is unrelated to reading a newspaper.
Given the context, the most appropriate answer is A) literacy, as it encompasses the skills that are practiced when reading a newspaper.
$\boxed{A}$

**Structured of Thoughts (SoT) Response:** *[Length: 69 characters - 94% shorter]*
`<think> #reading_newspaper → #improves → #literacy </think>` $\boxed{A}$

## D.5 TOKEN USAGE

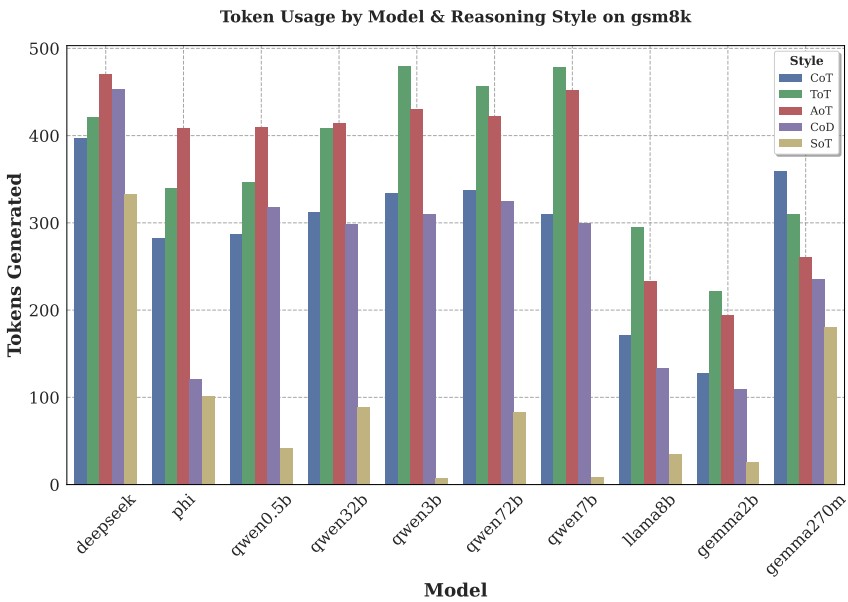

Figure 11: Token Usage on GSM8k

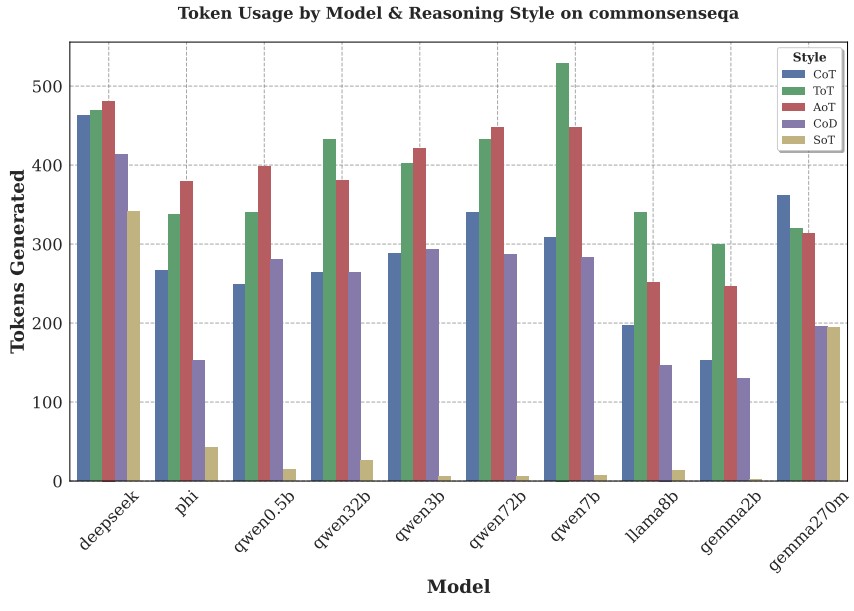

Figure 12: Token Usage on CommonsenseQA

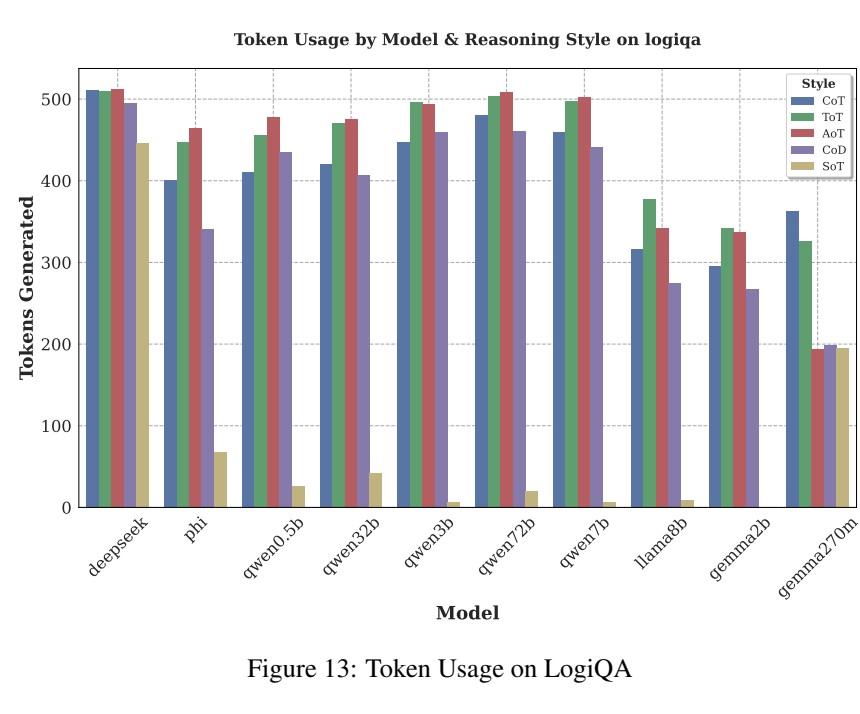

Figure 13: Token Usage on LogiQA

# E  CROSS-MODEL COMPARISON

## E.1  CASE STUDY 1: GAME24

Table 2: Final responses from different models and reasoning methods for Game24 problem "2, 2, 7, 10"

| Model | CoT | ToT | AoT | CoD | SoT |
|-------|-----|-----|-----|-----|-----|
| **Qwen 3B** | No solution ✓ | No solution ✓ | No solution ✓ | No answer × | No solution ✓ |
| **Qwen 7B** | No answer × | No answer × | No answer × | No answer × | No answer × |
| **Qwen 32B** | Invalid expr. × | No answer × | No answer × | No answer × | No answer × |
| **Qwen 72B** | **Valid sol.** ✓ | No solution ✓ | **Valid sol.** ✓ | Invalid expr. × | No answer × |
| **GPT-OSS 120B** | No answer × | No answer × | No answer × | No answer × | **Valid sol.** ✓ |

Table 3: Success rate analysis across models and methods

| Model | Valid Solutions | Correct "No Solution" | Invalid Solutions | No Answer | Success Rate |
|-------|-----------------|------------------------|-------------------|-----------|--------------|
| Qwen 3B | 0 | 4 | 0 | 1 | 80% |
| Qwen 7B | 0 | 0 | 0 | 5 | 0% |
| Qwen 32B | 0 | 0 | 1 | 4 | 0% |
| Qwen 72B | 2 | 1 | 1 | 1 | 60% |
| GPT-OSS 120B | 1 | 0 | 0 | 4 | 20% |
| **Total** | **3** | **5** | **2** | **15** | **32%** |

Table 4: Method performance across all models

| Method | Valid Solutions | Correct "No Solution" | Invalid Solutions | No Answer | Success Rate |
|--------|-----------------|------------------------|-------------------|-----------|--------------|
| CoT | 2 | 2 | 1 | 0 | 80% |
| ToT | 0 | 2 | 0 | 3 | 40% |
| AoT | 2 | 2 | 0 | 1 | 80% |
| CoD | 0 | 0 | 1 | 4 | 0% |
| SoT | 1 | 1 | 0 | 3 | 40% |

Table 5: Valid solutions discovered

| Model | Method | Expression and Verification |
|---|---|---|
| Qwen 72B | CoT | $(10 \div 2 + 7) \times 2 = (5 + 7) \times 2 = 12 \times 2 = 24$ |
| Qwen 72B | AoT | $(10 \div 2 + 7) \times 2 = (5 + 7) \times 2 = 12 \times 2 = 24$ |
| GPT-OSS 120B | SoT | $(7 + 10/2) \times 2 = (7 + 5) \times 2 = 12 \times 2 = 24$ |

Table 6: Detailed final expressions from each model-method combination

| Model | Method | Final Expression |
|---|---|---|
| Qwen 3B | CoT | No solution |
| | ToT | No solution |
| | AoT | No solution |
| | CoD | No final boxed answer (infinite loop) |
| | SoT | No solution |
| Qwen 7B | CoT | No final boxed answer (infinite loop) |
| | ToT | No final boxed answer (infinite loop) |
| | AoT | No final boxed answer (infinite loop) |
| | CoD | No final boxed answer (infinite loop) |
| | SoT | No final boxed answer (infinite loop) |
| Qwen 32B | CoT | $(10 + 2) \times 2 + 7 - 7$ (uses 7 twice) |
| | ToT | No final boxed answer (infinite loop) |
| | AoT | No final boxed answer (infinite loop) |
| | CoD | No final boxed answer (infinite loop) |
| | SoT | No final boxed answer (infinite loop) |
| Qwen 72B | CoT | $(10 \div 2 + 7) \times 2$ **(Valid solution)** |
| | ToT | No solution |
| | AoT | $(10 \div 2 + 7) \times 2$ **(Valid solution)** |
| | CoD | $(2 + 2) \times (10 - 7) + 10 + 2$ (number reuse) |
| | SoT | No final boxed answer (infinite loop) |
| GPT-OSS 120B | CoT | No final boxed answer (stream of consciousness) |
| | ToT | No final boxed answer (chaotic exploration) |
| | AoT | No final boxed answer (endless calculations) |
| | CoD | No final boxed answer (partial insight) |
| | SoT | $(7 + 10/2) \times 2$ **(Valid solution)** |

## E.2 Case Study 2: AIME

Let $n = 2^{31}3^{19}$. How many positive integer divisors of $n^2$ are less than $n$ but do not divide $n$?

**Ground Truth:** 589 **Legend:** Green = Correct (589), Orange = Near-correct (588), Red = Incorrect

| Model | CoT | ToT | AoT | CoD | SoT |
|---|---|---|---|---|---|
| Qwen3B | 1816 | 589 | 588 | 346 | 588 |

| Model | CoT | ToT | AoT | CoD | SoT |
|-------|-----|-----|-----|-----|-----|
| **Qwen7B** | 589 | 588 | 588 | 589 | 909 |
| **Qwen32B** | 588 | 588 | 588 | 588 | 588 |
| **Qwen72B** | 589 | 589 | 908 | 588 | 589 |
| **GPT-OSS-120B** | 589 | 589 | 589 | 589 | 908 |
| **Ground Truth** | 589 | | | | |

## F  EXPERIMENT SETTINGS

To ensure fair and reproducible comparisons across all models, we maintained consistent prompting configurations. We set the temperature to 0 for deterministic outputs and standardized both the maximum new tokens and model context length for each dataset.

Table 8: Prompting Configuration for All Models

| Dataset | Max New Tokens | Context Length | Task Characteristics |
|---------|----------------|----------------|----------------------|
| AIME | 2048 | 8192 | Complex mathematical reasoning |
| CommonsenseQA | 512 | 8192 | Multiple-choice questions |
| Game24 | 2048 | 8192 | Mathematical puzzle solving |
| GSM8K | 512 | 8192 | Grade school math problems |
| LogiQA | 512 | 8192 | Logical reasoning tasks |

## G  SFT EXPERIMENTAL SETUP (DATASET, TRAINING, AND EVALUATION)

For each problem in our dataset, we generate responses using five distinct reasoning style prompts applied to a capable base model. Each problem receives five different reasoning approaches while maintaining the same correct answer.

### G.1  SFT DATASET CONSTRUCTION PROCEDURE

**Style Selection Training Data** Our SFT approach trains models to automatically select the most appropriate reasoning style for each problem. The dataset consists of problems paired with optimal style choices determined through empirical evaluation.

### G.2  TRAINING DATA FORMAT

Each training example follows a conversational format with system instructions, user queries, and target style selections:

```
{
  "messages": [
    {
      "role": "system",
      "content": "Your task is to choose the most appropriate
                  reasoning style for answering the user's question.
                  You must choose from:
                  - CoT (Chain of Thought)
                  - CoD (Chain of Draft)
```

```
                         - ToT (Tree of Thought)
                         - SoT (Sketch of Thought)
                         - AoT (Algorithm of Thought)

                         The selection should follow two criteria:
                         1. The style must lead to the correct answer.
                         2. Among all styles that produce correct answers,
                            choose the one with the most concise response."
        },
        {
          "role": "user",
          "content": "[Problem statement with multiple choice options]"
        },
        {
          "role": "assistant",
          "content": "[Selected Style: CoT/CoD/ToT/SoT/AoT]"
        }
    ]
}
```

### G.3 STYLE SELECTION CRITERIA

The training data is constructed using a two-stage optimization process:

**Stage 1: Correctness Filtering** For each problem, we evaluate all five reasoning styles and identify which ones produce the correct answer.

**Stage 2: Conciseness Selection** Among the correct styles, we select the one with the most concise response based on:

- Token count
- Reasoning steps
- Computational complexity

This is the proportion of the predicted label in the training dataset.

### G.4 TRAINING CONFIGURATION

**Model Architecture:** We fine-tune base models from each scale category using the conversational format.

**Training Parameters:**

- Learning rate: $2 \times 10^{-5}$
- Batch size: 16
- Training epochs: 3
- Gradient clipping: 1.0
- Loss function: Cross-entropy loss on style classification

**Data Distribution:** The training set maintains balanced representation across:

- Problem types (math, reasoning, coding, puzzles)

- Difficulty levels

- Optimal style assignments

## G.5 EVALUATION PROTOCOL

**Style Selection Accuracy:** Measured as the percentage of problems where the model selects the empirically optimal style.

**Downstream Performance:** Evaluate whether automatic style selection maintains accuracy compared to human-selected styles.

This methodology enables models to automatically adapt their reasoning approach based on problem characteristics, potentially improving efficiency while maintaining accuracy across diverse tasks.

## G.6 TRAINING DATA DISTRIBUTION AND MODEL BEHAVIOR

**Ground Truth Style Distribution** Our training dataset of 3,000 problems exhibits an uneven distribution of optimal reasoning styles, as determined through empirical evaluation (Figure 14). At inference time, the fine-tuned model was prompted to first select the best reasoning style for a new question before applying that style to generate a solution. However, as shown in Figure 14, the SFT process failed to instill genuine strategic understanding. Instead, the model developed a strong bias towards consistently selecting Chain-of-Draft (CoD), regardless of the actual problem context. This pathological selection strategy effectively nullified any potential advantage over simply using a single, fixed style across all tasks.

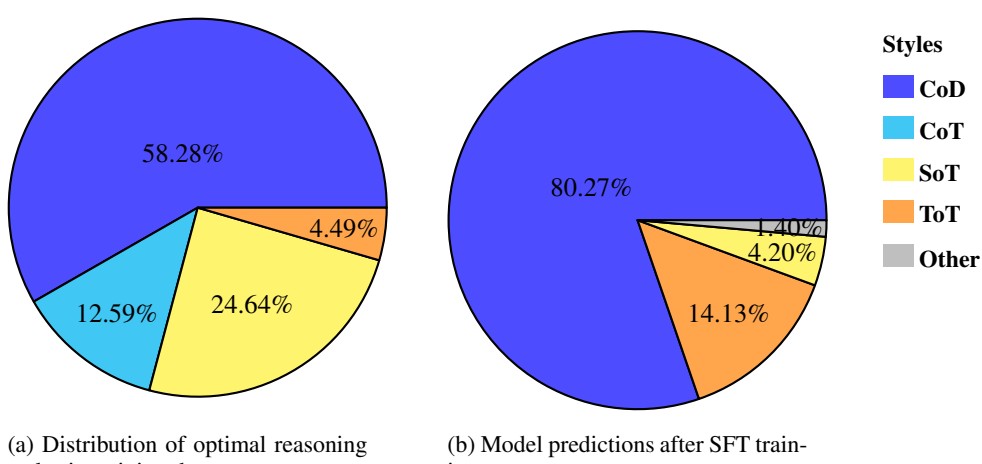

(a) Distribution of optimal reasoning styles in training data.

(b) Model predictions after SFT training.

Figure 14: Comparison of reasoning style distributions before and after training. CoD dominates both distributions, with increased prevalence after training.

The dominance of Chain-of-Draft (CoD) at 58.28% suggests that for most problems in our benchmark suite, a concise drafting approach provides the optimal balance between correctness and efficiency. Tree-of-Thought (ToT) represents only 4.49% of optimal solutions, indicating that multi-perspective reasoning is beneficial for a smaller subset of complex problems.

**Post-Training Model Predictions** After SFT, the model's prediction behavior shifts notably (Figure 14b), showing even stronger preference for CoD while developing capability to select ToT for appropriate problems.

| Reasoning Style | Count | Percentage |
|---|---|---|
| CoD (Chain of Draft) | 242 | 80.67% |
| ToT (Tree of Thought) | 44 | 14.67% |
| SoT (Sketch of Thought) | 11 | 3.67% |
| Other | 3 | 1.00% |
| **Total** | **300** | **100%** |

Table 9: Distribution of reasoning styles predicted by the fine-tuned model on 300 sampled questions.

The model demonstrates learned preference for CoD (80.67%) and increased selection of ToT (14.67%) compared to training distribution, suggesting the model has learned to identify problems where multi-perspective reasoning provides value.

### G.7    TRAINING DYNAMICS

We compare two fine-tuning approaches: Low-Rank Adaptation (LoRA) and full parameter fine-tuning on Qwen-7B.

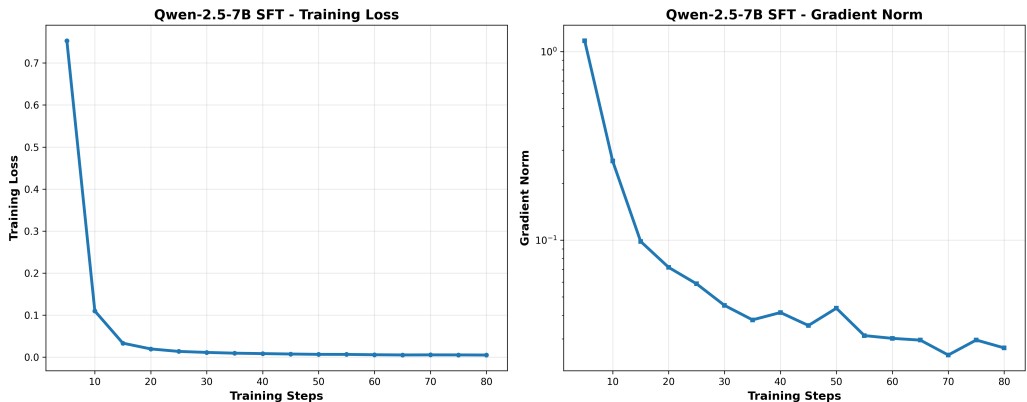

Figure 15: Training dynamics for Qwen-7B with LoRA fine-tuning. Left panel shows training loss convergence over steps. Right panel shows gradient norm evolution, indicating stable optimization throughout training.

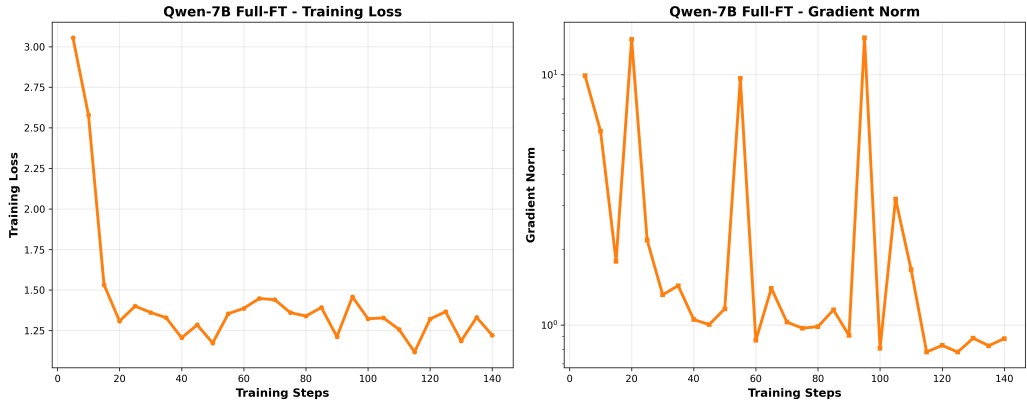

Figure 16: Training dynamics for Qwen-7B with full parameter fine-tuning. Left panel shows training loss convergence over steps. Right panel shows gradient norm evolution, demonstrating higher gradient magnitudes compared to LoRA fine-tuning.

Figures 15 and 16 demonstrate that both LoRA and full fine-tuning achieve stable convergence. LoRA fine-tuning exhibits more stable gradient norms, while full fine-tuning shows higher gradient magnitudes but maintains convergence, suggesting both approaches are viable for style selection training.

