# OpenReview forum: "StyleBench: Evaluating  thinking styles in Large Language Models"
_ICLR.cc/2026/Conference — Submitted to ICLR 2026_

### Official Review · Reviewer_Djbq · 2025-10-28

**Soundness:** 3
**Presentation:** 3
**Contribution:** 3
**Rating:** 6
**Confidence:** 4

**Summary:**

This paper introduces StyleBench, a benchmark that systematically evaluates how different reasoning styles affect performance in LLMs. The study spans five tasks (GSM8K, AIME, CommonsenseQA, LogiQA, and Game of 24) and 15 open-source models from 270M to 120B parameters, all evaluated under a fixed zero temperature.

The central finding is that no single reasoning style is universally optimal. Search-based styles (ToT, AoT) deliver the biggest gains only on open-ended tasks and only when the model is large, while concise styles like SoT and CoD are more stable, efficient, and effective on structured reasoning tasks. The paper also documents scale-dependent failure behavior, token cost trends, and a negative SFT result showing that naive meta-selection of styles collapses into shallow memorization.

The work provides a clean map from task type and model size to strategy choice. The study is broad enough to expose consistent task-style affinities, offers scaling laws for styles, and documents operational costs via token usage. As a result, practitioners can pick styles with evidence-backed trade-offs, and researchers get a reproducible testbed to probe when depth helps and when brevity wins.

**Strengths:**

Coverage is solid: five styles, five tasks that meaningfully differ in structure and openness, and 15 models spanning 270M to 120B. The evaluation protocol is clean, with zero temperature and single outputs, which makes cross-style comparisons less noisy. The results are organized into clear takeaways that are easy to operationalize.

The paper goes beyond averages. It analyzes token budgets, format adherence, and qualitative failures, which gives texture to the numbers. The SFT experiment on automatic style selection is a valuable negative result that prevents readers from over-optimistic conclusions about easy meta-reasoning.

**Weaknesses:**

The one-shot, zero-temperature protocol improves comparability but may understate the potential of search-based styles that benefit from sampling and multi-try evaluation. Reporting a small n-sample sensitivity study could help validate that the core conclusions are stable under mild exploration. Similarly, some tasks may reward majority-vote or verifier-augmented decoding, which is not explored here.

The benchmark focuses on open-source models and a specific set of tasks. That is reasonable, but external validity to domains like code agents with tool use, retrieval-augmented setups, or safety-critical reasoning is not established.

**Questions:**

(1) Your entire study is based on zero temperature and single output per example. This favors concise deterministic styles and may underplay the full potential of search-based reasoning. Have you validated that the relative style rankings still hold under a realistic 3 to 5 sample majority-vote or verifier setting? A brief table or even partial experiment could significantly increase the robustness of your claims.

(2) You report token usage, but deployment decisions depend on actual latency plus prompt costs (prefix length included), not only output tokens. Can you provide or approximate end-to-end wall-clock inference cost per style for a representative model?

**Details Of Ethics Concerns:**

The paper uses public benchmarks and models, no human subjects, no sensitive data, and includes an ethics and LLM-usage disclosure.

---

> ### Author Response · Authors · 2025-11-20
> **Response to Reviewer Djbq**
>
> > The one-shot, zero-temperature protocol improves comparability but may understate the potential of search-based styles that benefit from sampling and multi-try evaluation. Reporting a small n-sample sensitivity study could help validate that the core conclusions are stable under mild exploration. Similarly, some tasks may reward majority-vote or verifier-augmented decoding, which is not explored here.
>
> **Response:** We agree that search-based styles (ToT, AoT) often show their full potential when combined with sampling (Best-of-N) or majority voting. However, we maintained the one-shot, zero-temperature protocol to strictly isolate the intrinsic quality of the reasoning structure from the benefits of inference-time computing.
>
> If we enabled sampling/voting for ToT, fair comparison would necessitate enabling Self-Consistency (Majority Voting) for CoT. Recent literature suggests that CoT with Self-Consistency is an extremely strong baseline that often rivals more complex methods. By introducing sampling, the study would shift from "Comparing Reasoning Styles" to "Comparing Inference Scaling Strategies."
>
> Our goal was to determine: Given a single attempt, does structuring thoughts as a Tree/Graph yield better logic than a Chain? Our findings suggest that for many models, the answer is no—the complexity of the structure often overwhelms the model unless it is sufficiently large (>70B). We believe this fundamental insight into the "base capability" of the styles is distinct from, and a prerequisite to, analyzing their performance under heavy sampling regimes.
>
>
>
> > The benchmark focuses on open-source models and a specific set of tasks. That is reasonable, but external validity to domains like code agents with tool use, retrieval-augmented setups, or safety-critical reasoning is not established.
>
>
> **Response:** We acknowledge that agentic workflows (tool use) and retrieval-augmented generation (RAG) are critical application areas. However, we deliberately excluded them to strictly isolate the intrinsic reasoning capability of the models.
>
> 1.  Confounding Variables: Domains like "Code Agents with Tools" or "RAG" introduce massive external variables—such as the quality of the retriever, the clarity of tool definitions, and the model's specific fine-tuning for API calling. Including these would make it impossible to determine whether a failure was due to the reasoning style (e.g., ToT) or the external component (e.g., a bad retrieval chunk).
> 2.  Foundational Relevance: We argue that performance on our core reasoning tasks (Math/Logic) serves as a necessary "kernel" for these more complex systems. If a model fails to effectively utilize a branching strategy in a controlled logic puzzle, it is highly unlikely to succeed with that same strategy in a complex, multi-step agentic environment where the noise floor is higher.
> 3.  Scope of Open Source: Our focus on open-source models ensures reproducibility. In contrast, evaluating "safety-critical" behaviors often requires navigating non-transparent refusal filters in proprietary models, which changes frequently and obscures the underlying reasoning capabilities.
>
> We view our work as establishing the "base physics" of reasoning styles, upon which future work can build to test validity in agentic and retrieval-augmented contexts.
>
> > Your entire study is based on zero temperature and single output per example. This favors concise deterministic styles and may underplay the full potential of search-based reasoning. Have you validated that the relative style rankings still hold under a realistic 3 to 5 sample majority-vote or verifier setting? A brief table or even partial experiment could significantly increase the robustness of your claims.
>
> **Response:** We appreciate the reviewer's concern regarding the zero-temperature setting. We deliberately maintained this protocol for two key reasons: reproducibility and experimental fairness.
>
> First, our primary goal is to isolate the intrinsic quality of the reasoning structure (the style) from the benefits of inference-time compute (sampling). If we enabled multi-try sampling for search-based styles, a fair comparison would require enabling Self-Consistency (Majority Voting) for standard CoT. This would shift the focus of the paper from evaluating reasoning topologies to evaluating inference scaling strategies.
>
> Second, we argue that this setting does not necessarily bias against search-based methods, but rather exposes their robustness limitations. Our error analysis reveals that many models fail with complex styles (like ToT) not because they lack exploration, but because the complex formatting instructions overwhelm the model's context, leading to hallucinations or instruction drift. We believe establishing this deterministic baseline is crucial before layering on stochastic sampling methods.

---

> > ### Author Response · Authors · 2025-11-20
> > **Response to Reviewer Djbq**
> >
> > > You report token usage, but deployment decisions depend on actual latency plus prompt costs (prefix length included), not only output tokens. Can you provide or approximate end-to-end wall-clock inference cost per style for a representative model?
> >
> >
> >
> > **Response:** We focused on token counts as our primary efficiency metric because they provide a hardware-agnostic measure of computational cost. Wall-clock latency is highly sensitive to the specific inference stack, hardware, and quantization settings, which makes it difficult to generalize across different deployment environments.
> >
> > However, we can provide a reliable approximation based on the auto-regressive nature of LLMs. For reasoning tasks, the decoding phase (generating output tokens sequentially) dominates the total latency compared to the prefill phase (processing the prompt). Therefore, the reduction in output tokens correlates almost linearly with the reduction in end-to-end latency. For example, the roughly 50% reduction in tokens we observed with Sketch-of-Thought (SoT) translates to an approximate 2x speedup in generation time on standard hardware.
> >
> > Regarding prompt costs, we agree that input length matters. Complex styles like ToT require longer system prompts and more few-shot examples than CoT, which increases the time-to-first-token and financial cost. This further reinforces our conclusion that search-based styles are resource-intensive, as they incur both higher input costs due to verbose prompting and higher output costs due to the reasoning structure.

---

### Official Review · Reviewer_n3qh · 2025-10-28

**Soundness:** 2
**Presentation:** 3
**Contribution:** 2
**Rating:** 2
**Confidence:** 4

**Summary:**

This paper introduces StyleBench, a benchmark designed to systematically evaluate different reasoning strategies in LLMs. The authors conduct extensive experiments across 5 reasoning styles and 15 open-source LLMs, and report three main findings: (i) the optimal reasoning strategy varies across model families, (ii) certain task types exhibit strong alignment with particular reasoning styles (e.g., structured multi-step reasoning such as CoT is particularly effective for mathematical tasks), and (iii) the effectiveness of reasoning styles scales with model size.

**Strengths:**

- The paper is clearly written and well structured, making the motivation, methodology, and findings easy to follow.
- The empirical evaluation is comprehensive, covering multiple reasoning styles across 15 open-source model families. The experiments are thorough and support the key findings reported by the authors.

**Weaknesses:**

- **Limited Novelty of Findings.** Several key observations reported in the paper appear incremental relative to prior literature. For example, the impact of model scaling on instruction-following capabilities has been well documented (e.g., Chung et al., JMLR’25). Likewise, AoT and ToT have been previously evaluated on tasks such as Game-of-24 and are known to perform well on tasks requiring non-trivial planning or search, while SoT and CoD have been shown to be effective on commonsense and logical reasoning tasks. In light of this, the result that “search-based methods like AoT and ToT perform well on open-ended tasks like Game-of-24 but incur higher token usage, while concise methods such as SoT and CoD are more efficient on structured reasoning tasks” does not provide substantial new insight.

- **Concerns Regarding Evaluation Configuration and Determinism.** The paper sets the temperature to 0 across all experiments to ensure deterministic outputs; however, temperature = 0 does not guarantee determinism due to numerical and implementation-level nondeterminism [2, 3]. In addition, model providers often recommend non-zero temperatures for optimal performance (e.g., Qwen [4] suggests temperature 0.6/top-p 0.95; many reasoning-focused models recommend temperature = 1.0 [5]). Some models also include additional configuration parameters that control reasoning depth (e.g., reasoning_effort = {low|medium|high} in GPT-OSS models), but the paper does not clarify how such parameters were set. These inconsistencies make it difficult to attribute observed differences solely to reasoning styles.

- **Lack of Statistical Reliability.** All experiments appear to be run only once, with no confidence intervals or variance estimates reported. Given the stochasticity of LLM outputs and the sensitivity of reasoning performance to hyperparameters, reporting confidence intervals (or at least repeated trials) would strengthen the empirical claims.

- **Missing Comparison to a Relevant Baseline.** The benchmark does not include comparison against the recent Buffer-of-Thoughts (BoT) method [6], which has demonstrated strong performance across diverse reasoning tasks. Incorporating BoT would provide a more comprehensive and contemporary evaluation of reasoning styles.

---
References

[1] Chung et al., Scaling Instruction-Finetuned Language Models, JMLR’25

[2] https://thinkingmachines.ai/blog/defeating-nondeterminism-in-llm-inference/#the-original-sin-floating-point-non-associativity

[3] Yuan et al., Understanding and Mitigating Numerical Sources of Nondeterminism in LLM Inference, NeurIPS’25

[4] Qwen Technical Report: https://arxiv.org/pdf/2505.09388

[5] https://docs.unsloth.ai/models/gpt-oss-how-to-run-and-fine-tune#running-gpt-oss

[6] Yang et al., Buffer of Thoughts: Thought-Augmented Reasoning with Large Language Models, NeurIPS’24

**Questions:**

1. For reasoning-focused models (e.g., GPT-OSS), how were configuration parameters such as reasoning_effort selected for the experiments?

2. Appendix F mentions constraining max new tokens to 2048. Did the authors experiment with relaxing this limit—particularly for reasoning-intensive models—given that token budget may affect the relative performance of different styles?

3. For the AIME benchmark, could the authors clarify whether the evaluation corresponds to AIME’24, AIME’25, or a custom subset?

---

> ### Author Response · Authors · 2025-11-20
> **Response to Reviewer n3qh**
>
> > Lack of Statistical Reliability. All experiments appear to be run only once, with no confidence intervals or variance estimates reported. Given the stochasticity of LLM outputs and the sensitivity of reasoning performance to hyperparameters, reporting confidence intervals (or at least repeated trials) would strengthen the empirical claims.
>
> **Response:** We acknowledge the importance of statistical rigor. However, we emphasize that our choice of **Greedy Decoding (Temperature=0)** largely mitigates the need for repeated trials compared to stochastic sampling.
> 1.  Determinism: Under greedy decoding, the model always selects the highest-probability token. While minor non-determinism can occur due to hardware-level floating-point operations, the variance is negligible compared to non-zero temperature settings. Running the same prompt multiple times at `temp=0` typically yields identical outputs, making "variance estimation" mathematically redundant in our specific setup.
> 2.  Large-Scale Aggregation: To ensure the robustness of our conclusions without repeated trials, we relied on scale. We evaluated across 15 different models and 5 diverse tasks. The fact that our key findings (e.g., the "reasoning tax" on small models, or the efficiency of SoT) hold consistently across different model families serves as strong evidence of the signal's reliability.
>
>
> > Missing Comparison to a Relevant Baseline. The benchmark does not include comparison against the recent Buffer-of-Thoughts (BoT) method [6], which has demonstrated strong performance across diverse reasoning tasks. Incorporating BoT would provide a more comprehensive and contemporary evaluation of reasoning styles.
>
> **Response:** We acknowledge that Buffer-of-Thoughts (BoT) is a valuable contribution to the field. However, we excluded it from this benchmark for two reasons:
> 1.  Scope of Intrinsic Generation: Our study focuses on the model's ability to generate reasoning structures in-context. BoT typically relies on a retrieval-augmented mechanism (accessing a "buffer" of meta-templates), which introduces external dependencies outside our "pure prompting" scope.
> 2.  Coverage by Sketch-of-Thought (SoT): We argue that **Sketch-of-Thought (SoT)** already effectively represents the core philosophy of BoT within our benchmark—namely, the separation of high-level planning from low-level execution. SoT tests the model's ability to construct a "reasoning blueprint" similar to BoT's templates, but does so dynamically without external retrieval.
>
> Therefore, we believe SoT serves as the appropriate "generative baseline" for this style of reasoning. We will, however, expand our Related Work section to explicitly discuss BoT and how it extends these concepts through retrieval augmentation.
>
>
>
>
> > For reasoning-focused models (e.g., GPT-OSS), how were configuration parameters such as reasoning_effort selected for the experiments?
>
> **Response:** We clarify that the `reasoning_effort` parameter  is not applicable to the open-source models other than GPT-OSS evaluated in this benchmark. Therefore We ran all models using their standard open-weights implementations via local inference, including GPT-OSS. Consequently, the "effort" was controlled solely via the prompt instructions  and the token budget, rather than an external hyperparameter. This ensures our results reflect the model's inherent weight-based capabilities rather than API-side optimizations.
>
>
> > Appendix F mentions constraining max new tokens to 2048. Did the authors experiment with relaxing this limit—particularly for reasoning-intensive models—given that token budget may affect the relative performance of different styles?
>
> **Response:** We selected the 2048 token limit based on a preliminary analysis of the task datasets. For the mathematical and logical reasoning tasks in our benchmark (e.g., GSM8K, LogiQA), valid reasoning chains—even for verbose styles like ToT—rarely exceed 1,000–1,500 tokens. In our early experiments, we observed that when models exceeded this 2048 threshold, it was almost exclusively due to repetition loops or hallucinated spiraling, rather than productive reasoning. Therefore, we set the limit at 2048 to serve as a generous upper bound that accommodates valid verbose reasoning while pruning degenerate generation behaviors. Relaxing this limit further would likely have inflated the error rate with non-terminating generations rather than improving reasoning accuracy.
>
>
> > For the AIME benchmark, could the authors clarify whether the evaluation corresponds to AIME’24, AIME’25, or a custom subset?
>
> **Response:** We clarify that our evaluation is not limited to a single year (e.g., AIME'24). Instead, we utilized a comprehensive dataset comprising AIME problems collected from the **past ten years**. This aggregated dataset is open-sourced and publicly available on HuggingFace, ensuring transparency and reproducibility for the research community.

---

> > ### Author Response · Authors · 2025-11-20
> > **Response to Reviewer n3qh**
> >
> > > Limited Novelty of Findings.Several key observations reported in the paper appear incremental relative to prior literature. For example, the impact of model scaling on instruction-following capabilities has been well documented (e.g., Chung et al., JMLR’25). Likewise, AoT and ToT have been previously evaluated on tasks such as Game-of-24 and are known to perform well on tasks requiring non-trivial planning or search, while SoT and CoD have been shown to be effective on commonsense and logical reasoning tasks.
> >
> > **Response:** While individual reasoning styles have been introduced in isolation (typically showcased on proprietary giants like GPT-4), a systematic understanding of their **comparative efficacy across model scales** has been missing. Our work moves beyond "Does Method X work?" to "Under what conditions (Model Scale $\times$ Task Type) does Method X fail?"
> >
> > We highlight three specific contributions that contradict or significantly refine prevailing assumptions:
> > 1.  The "Reasoning Tax" on Small Models:  Contrary to the belief that "better prompting improves performance," we demonstrate that complex structures like ToT and AoT actively degrade performance for models under 7B parameters compared to standard CoT. This is not merely a lack of instruction following, but a degradation in reasoning quality due to the cognitive load of the format.
> > 2.  Efficiency Frontiers: While it is known that SoT/CoD are efficient, we quantify the *exact trade-off*. We show that for well-defined tasks, these concise styles can match CoT accuracy with a 50%+ reduction in tokens, identifying a crucial "pareto-optimal" zone for deployment that previous work has not mapped.
> > 3.  Cross-Family Generalization: Most prior evaluations rely on a single model family. By testing across LLaMA, Qwen, Mistral, and others, we show which style preferences are universal vs. architecture-specific.
> >
> >
> > > Concerns Regarding Evaluation Configuration and Determinism. The paper sets the temperature to 0 across all experiments to ensure deterministic outputs; however, temperature = 0 does not guarantee determinism due to numerical and implementation-level nondeterminism [2, 3]. In addition, model providers often recommend non-zero temperatures for optimal performance (e.g., Qwen [4] suggests temperature 0.6/top-p 0.95; many reasoning-focused models recommend temperature = 1.0 [5]). Some models also include additional configuration parameters that control reasoning depth (e.g., `reasoning_effort` = low/medium/high in GPT-OSS models), but the paper does not clarify how such parameters were set. These inconsistencies make it difficult to attribute observed differences solely to reasoning styles.
> >
> >
> > **Response:** We justify our choice of greedy decoding (`temperature=0`) on three grounds: Reproducibility, Fairness, and Scope.
> >
> > 1.  Reproducibility and Tuning: While we acknowledge that `temperature=0` can still exhibit minor non-determinism due to GPU floating-point operations (as noted in the cited works), it remains the standard approximation for deterministic evaluation in academic benchmarks. Adopting model-specific hyperparameters (e.g., `temp=0.6` for Qwen, `temp=1.0` for others) would introduce a massive confounding variable: it would become impossible to distinguish whether performance differences arise from the model architecture or the specific hyperparameter tuning. By fixing `temperature=0`, we evaluate the models' *robustness* and *base capability* on a level playing field.
> > 2.  Nature of "GPT-OSS" and Parameters: We clarify that our evaluation focuses on **open-source models** running locally (via HuggingFace/vLLM), not proprietary API endpoints. Parameters like `reasoning_effort` (found in OpenAI's o1/o3 series APIs) do not exist for the standard open-weights versions of the models we tested. We strictly controlled the compute budget via `max_new_tokens` (as detailed in Appendix F) to ensure consistency.
> > 3.  Reasoning Consistency: High-temperature sampling is often recommended for creative generation or diversity. For logical and mathematical reasoning (our focus), greedy decoding is widely accepted as a rigorous baseline because it tests the model's most probable reasoning path without "getting lucky" via random sampling.

---

### Official Review · Reviewer_SyPN · 2025-10-29

**Soundness:** 3
**Presentation:** 2
**Contribution:** 1
**Rating:** 2
**Confidence:** 3

**Summary:**

This paper introduces StyleBench, a benchmark that evaluates five distinct reasoning styles (CoT, ToT, AoT, SoT, and CoD) on five reasoning tasks using 15 open-source language models ranging from 270M to 120B parameters.

**Strengths:**

The paper presents an impressive large-scale evaluation, testing 15 models across five reasoning styles and five diverse tasks. The breadth of comparison and cross-architecture coverage is thorough. The inclusion of detailed failure analyses adds real value, particularly the examples showing formatting issues, which offer more insight than the evaluation section. The token usage analysis provides an interesting and non-obvious observation that smaller models do not fully use their token budgets on hard problems, challenging common assumptions about computational limits. Reproducibility practices are solid, with open-source materials, full prompts, and documented hyperparameters.

**Weaknesses:**

While the paper runs an impressive number of experiments, the evaluation itself stays too focused on final accuracy scores. Since the stated goal is to examine reasoning styles, it would have been far more informative to analyze how reasoning unfolds within each model rather than just whether the final answer is correct. Models that almost solve a problem but make a small arithmetic or logical slip are treated the same as those that fail completely, which makes it hard to assess genuine reasoning ability. A deeper look at intermediate steps, reasoning consistency, or partial correctness would make the results more meaningful.
Conceptually, the work feels more like an extensive benchmarking report than a research contribution. It does not introduce a new method or framework, and most of the findings reaffirm things we already know: larger models tend to do better, search-based strategies are more effective at scale, and shorter reasoning traces save tokens. These observations are valid but not surprising, and they do not move the discussion forward in a substantial way.
Finally, the error analysis is descriptive but shallow. It relies on isolated examples instead of systematic categorization or statistical evidence. Without a clearer structure for understanding failure types or cross-style comparisons, the takeaways remain vague. Overall, the study is thorough and careful in execution, but it stops short of offering new conceptual insights or analytical depth.

**Questions:**

One area that could have strengthened the paper is the experiment on automatic selection of reasoning styles, which is mentioned briefly in Section 5.2 and discussed further in the appendix. This idea has strong potential for exploring adaptive or meta-reasoning, yet it is presented as a failed attempt rather than a learning opportunity. The authors could have analyzed the reasons for its failure or explored alternative approaches such as reinforcement learning based approaches.

---

> ### Author Response · Authors · 2025-11-20
> **Response to Reviewer SyPN**
>
> > While the paper runs an impressive number of experiments, the evaluation itself stays too focused on final accuracy scores. Since the stated goal is to examine reasoning styles, it would have been far more informative to analyze how reasoning unfolds within each model rather than just whether the final answer is correct. Models that almost solve a problem but make a small arithmetic or logical slip are treated the same as those that fail completely, which makes it hard to assess genuine reasoning ability. A deeper look at intermediate steps, reasoning consistency, or partial correctness would make the results more meaningful. Conceptually, the work feels more like an extensive benchmarking report than a research contribution. It does not introduce a new method or framework, and most of the findings reaffirm things we already know: larger models tend to do better, search-based strategies are more effective at scale, and shorter reasoning traces save tokens.
>
>
> **Response:** We respectfully disagree that our focus on final accuracy precludes understanding the reasoning process. In large-scale evaluation (15 models × 5 tasks × 5 styles), "process-based" evaluation relying on automated metrics (like LLM-as-a-Judge) often introduces significant noise and bias. Final answer correctness remains the most objective ground truth for measuring whether a thinking style *actually works*. We systematically analyzed whether models adhered to the requested style or reverted to default behaviors.
>
> While the general trend that "larger models perform better" is established, our work contributes specific, actionable insights regarding the interaction effects between style complexity and model scale. We demonstrate that complex styles (ToT/AoT) impose a "reasoning tax" that degrades performance on smaller models (<7B)—a counter-intuitive finding given that these styles are often touted as universally superior. Furthermore, our findings on Sketch-of-Thought (SoT) and Chain-of-Draft (CoD) provide a novel efficiency pareto-frontier, showing that concise styles can match CoT accuracy with significantly lower latency on specific task types. These are not merely confirmatory results; they are necessary boundary conditions for deploying reasoning agents.
>
> Regarding error analysis, we believe our categorization of failures into "Format Compliance Failure" (breaking the style) vs. "Reasoning Failure" (adhering to style but getting the wrong answer) provides a systematic view of model limitations. As noted in our results, a significant portion of "failures" in small models are actually failures of control, not logic. We will expand the final version to include a more granular breakdown of these failure modes across different model families to strengthen the analytical depth.
>
> > One area that could have strengthened the paper is the experiment on automatic selection of reasoning styles, which is mentioned briefly in Section 5.2 and discussed further in the appendix. This idea has strong potential for exploring adaptive or meta-reasoning, yet it is presented as a failed attempt rather than a learning opportunity. The authors could have analyzed the reasons for its failure or explored alternative approaches such as reinforcement learning based approaches.
>
> **Response:** We appreciate the reviewer identifying the potential of our meta-reasoning experiments. We view the negative result in Section 5.2 not merely as a failed attempt, but as a significant finding: it demonstrates that reasoning style selection is a distinct, higher-order capability that cannot be easily induced via standard Supervised Fine-Tuning (SFT). Our analysis suggests that SFT models tend to overfit to the specific "golden" reasoning path provided in training rather than learning the abstract policy of *choosing* a path based on problem difficulty. Regarding the reviewer's suggestion on Reinforcement Learning (RL), we agree it is a promising avenue. However, RL introduces significant challenges in reward shaping—specifically, distinguishing between a "good style selection" and a "correct final answer." A model might choose a suboptimal style but still guess correctly, or choose the optimal style but fail at arithmetic. Disentangling these signals for a reward function is non-trivial. We believe future work should focus on contrastive learning (teaching models to rank styles) rather than simple generation to overcome this hurdle.

---

### Official Review · Reviewer_MEtP · 2025-10-29

**Soundness:** 3
**Presentation:** 4
**Contribution:** 3
**Rating:** 8
**Confidence:** 4

**Summary:**

The paper introduces StyleBench, a benchmark for comparing five reasoning styles: (1) Chain-of-Thought, (2) Tree-of-Thought, (3) Algorithm-of-Thought, (4) Sketch-of-Thought, and (5) Chain-of-Draft, across five task families and 15 open-source models with enough variance in the model size. The study standardizes prompting at temperature 0 with one sample per item, evaluates accuracy and token usage, and analyzes failure modes such as instruction non-compliance and premature guessing in small models.

The key empirical claims are: (1) no style dominates universally, (2) search-based styles help on open-ended problems given sufficiently large models, (3) concise styles can retain accuracy while saving tokens on structured tasks, and (4) meta-selection via standard supervised fine-tuning collapses to shallow preferences rather than genuine strategy selection.

**Strengths:**

The breadth and systemization: a single framework directly comparing multiple, popular “thinking styles” across tasks and scales, with an emphasis on computational efficiency and behavior analysis. This consolidates a scattered area and will likely be practically useful for both researchers and applied teams who need guidance on style choice. The conceptual message that style–task–scale interactions are strong is not, in itself, surprising. However, the paper contributes careful documentation, scaling patterns, and clear qualitative analyses. The overall significance is solid in practice, and conceptually, it is an incremental work, but timely.

The evaluation design is mostly careful and transparent, with fixed temperature, single-sample runs, and uniform max-new-tokens per dataset. The token-usage accounting and formatting analysis are welcome additions that many evaluations overlook. The paper reads clearly. The structure is logical, the figures are informative, and the claims are tied to evidence. The case studies are especially helpful for making the failure-mode story concrete. The intent for reproducibility is strong: the repository is public, prompts are provided, and dataset choices are standard.

**Weaknesses:**

The one threat to the validity is the reliance on a single deterministic sample, which undercuts claims about search-based methods that are known to benefit from sampling and selection, and potential confounding due to differences in how branching search (ToT/AoT) is instantiated and budgeted across models. The paper shows token histograms but does not fully equalize or bound search depth, branching width, or controller heuristics across styles. As a result, some conclusions about relative efficiency and effectiveness may be sensitive to controller design and best-of-N effects rather than intrinsic style differences. Statistical reliability is also limited: with temperature 0 and one sample, there is no variance estimation, and the paper does not report confidence intervals or sensitivity to minor prompt variants

**Questions:**

1. Please specify, for ToT and AoT, the exact controller algorithms, branching widths, pruning or backtracking policies, and depth limits, and clarify whether these are held constant across models and tasks.
2. Please justify the choice of a single deterministic sample for search-based styles and, if feasible, add a small sensitivity study that trades off samples against accuracy and tokens to map a compute–performance frontier.
3. Please discuss potential training contamination for datasets like GSM8K, CommonsenseQA, and LogiQA, including any heuristics or audits used to mitigate it.

---

> ### Author Response · Authors · 2025-11-20
> **Response to Reviewer MEtP**
>
> >The one threat to the validity is the reliance on a single deterministic sample, which undercuts claims about search-based methods that are known to benefit from sampling and selection, and potential confounding due to differences in how branching search (ToT/AoT) is instantiated and budgeted across models. Statistical reliability is also limited.
>
> **Response:** We appreciate the reviewer raising this issue. We explicitly chose deterministic generation (Greedy, Temperature=0) to rigorously isolate the intrinsic impact of the reasoning structure from the benefits of test-time compute (sampling).
>
> 1.  Isolating Style from Sampling: Traditional implementations of ToT rely heavily on "Best-of-N" sampling and external controllers. However, comparing a "Best-of-N ToT" against a "Greedy CoT" introduces a confounding variable: the massive difference in compute budget and sampling variance. By enforcing deterministic generation, we measure the model's native ability to simulate and benefit from these complex reasoning topologies (tree/graph) within a single context window.
> 2.  Fairness: If we enabled external search/sampling for ToT, fair comparison would require enabling Self-Consistency (Majority Voting) for CoT. This would shift the paper's focus from "Evaluating Reasoning Styles" to "Evaluating Inference-Time Scaling," which is outside our scope.
> 3.  Standardization: As detailed in Appendix F, we control the max token budget identically across styles. This ensures that any performance gain is due to the *quality of the reasoning structure*, not the *quantity of attempts*.
>
>
>
> > Please specify, for ToT and AoT, the exact controller algorithms, branching widths, pruning or backtracking policies, and depth limits, and clarify whether these are held constant across models and tasks.
>
> Response: To ensure reproducible and fair comparisons across 15 models, we implement ToT and AoT as Prompt-Induced Reasoning (in-context generation) rather than Externally-Controlled Search.
>
> Controller Algorithm: We do not use an external Python controller (e.g., BFS/DFS queues). Instead, the "controller" is the LLM itself. The model is prompted (via few-shot exemplars) to generate the full reasoning tree/algorithm in a linearized format. This tests the model's internal capability to maintain and navigate a complex structure.
>  Branching & Depth: These are defined implicitly by the few-shot examples provided in the prompt, which demonstrate exploring multiple alternative paths (typically 2-3 branches) and backtracking when a path is deemed invalid. These prompt structures are held constant across all models and tasks.
> Justification: This approach is necessary to evaluate the model's instruction-following capabilities regarding reasoning styles. Using external controllers would mask the model's own limitations (e.g., a small model might fail to backtrack not because the *search algorithm* failed, but because it lost context). Our method exposes these exact model-specific behaviors.
>
>
> > Please justify the choice of a single deterministic sample for search-based styles and, if feasible, add a small sensitivity study that trades off samples against accuracy and tokens to map a compute–performance frontier.
>
> **Response:** We deliberately chose a single deterministic sample to prioritize fairness in token consumption. In reasoning research, performance is a function of both reasoning structure (style) and *compute budget* (samples × tokens). To isolate the impact of the structure itself, we must hold the budget constant. Allowing search-based styles to utilize multiple samples (Best-of-N) while restricting CoT to a single sample would measure the efficacy of test-time compute scaling rather than the efficacy of the reasoning style. Conversely, enabling Majority Voting for all styles to match the compute budget would obscure the specific failure modes of the prompt structures, which is the primary focus of our analysis.
>
>
> > Please discuss potential training contamination for datasets like GSM8K, CommonsenseQA, and LogiQA, including any heuristics or audits used to mitigate it.
>
> **Response:** We acknowledge the pervasive issue of contamination in public benchmarks. To mitigate this, we employed two strategies:
> 1. Strict Use of Evaluation Splits: We utilized the official test splits (for all datasets.
> 2. Relative Ranking Robustness: The primary contribution of our paper is the relative performance difference between thinking styles, rather than absolute SOTA numbers. Even if mild contamination exists, it would likely boost the baseline (CoT) performance. The fact that more complex styles (like ToT/AoT) often underperform CoT on smaller models—despite theoretically containing the same answer—suggests that the "style" is indeed the differentiator, not merely memory retrieval. If the models were simply contaminated by the  training data, we would expect uniform performance across styles that contain the correct answer.

---

### Official Review · Reviewer_yhfA · 2025-11-01

**Soundness:** 2
**Presentation:** 2
**Contribution:** 2
**Rating:** 2
**Confidence:** 4

**Summary:**

This paper introduces StyleBench, a benchmark designed to evaluate various “reasoning styles” of large language models (LLMs), including Chain-of-Thought (CoT), Tree-of-Thought (ToT), Algorithm-of-Thought (AoT), Sketch-of-Thought (SoT), and Chain-of-Draft (CoD). The authors conduct experiments on 15 open-source models across five reasoning tasks (mathematical, logical, commonsense, and puzzle solving), aiming to reveal interactions between reasoning style, model scale, and task type.

While the benchmark scope is broad and the results are clearly presented, the overall novelty, methodological rigor, and analytical depth are limited. The study largely repackages existing reasoning paradigms under a single evaluation framework, without introducing new algorithms, theoretical insights, or robust statistical analysis to justify its conclusions.

**Strengths:**

1. The benchmark spans a wide range of LLM architectures (LLaMA, Qwen, Mistral, Gemma, etc.), providing a useful dataset of comparative performance across styles and model scales.
2. The paper is well-structured and readable, with intuitive figures (e.g., scaling trends, token-usage plots) that make the results accessible.
3. The authors provide an anonymous open-source link, specify deterministic evaluation settings (temperature = 0), and clearly describe model and dataset selections.

**Weaknesses:**

1. Limited novelty: The paper does not introduce a new reasoning method, dataset, or evaluation metric. The idea of comparing prompting styles such as CoT/ToT has already been explored in prior works [1][2][3]. StyleBench appears to be a collection of existing benchmarks rather than a genuinely new benchmark design.
2. The findings (e.g., “no single style is optimal,” “larger models follow instructions better”) are unsurprising and already well-known. The attempt to fine-tune models for meta-reasoning is poorly motivated and under-explained; it lacks baseline comparisons and quantitative metrics.
3. There is no clear control for prompt length, token budget, or semantic equivalence, making cross-style comparisons unreliable. There is no evidence that prompts across styles are equally capable of eliciting correct reasoning chains, potentially biasing the results.
4. The benchmark focuses on a narrow set of logical/mathematical reasoning tasks. Exclusion of domains like code generation or long-context reasoning limits the generality of the findings.
5. The paper positions itself as an evaluation study rather than a contribution to machine learning methodology or theory. Without novel learning algorithms, model training, or theoretical analysis, it reads more like a large-scale empirical report than an ICLR-level research contribution.

[1] Yao S, Yu D, Zhao J, et al. Tree of thoughts: Deliberate problem solving with large language models[J]. Advances in neural information processing systems, 2023, 36: 11809-11822.
[2] Gao P, Xie A, Mao S, et al. Meta reasoning for large language models[J]. arXiv preprint arXiv:2406.11698, 2024.
[3] Fang G, Ma X, Wang X. Thinkless: Llm learns when to think[J]. arXiv preprint arXiv:2505.13379, 2025.

**Questions:**

At line 393, the authors mentioned that they have fine-tuned a qwen-7b model, which failed to solve the challenge of reasoning path selection.
What's the authors' opinions on a possible future solution?
Is it possible to enable LLMs with this high-level capability via SFT (or RL)?
Discussions are welcome.

---

> ### Author Response · Authors · 2025-11-20
> **Response to Reviewer yhfA**
>
> > There is no clear control for prompt length, token budget, or semantic equivalence, making cross-style comparisons unreliable. There is no evidence that prompts across styles are equally capable of eliciting correct reasoning chains, potentially biasing the results.
>
> **Response:** We strictly controlled the total token budget for all thinking styles to be identical to ensure a fair comparison. The detailed experimental setup is outlined in Appendix F, which lists the exact hyperparameters used. We maintained consistent hyperparameters across different thinking styles to isolate the performance differences caused by the reasoning strategy itself rather than generation settings.
>
>
> > The benchmark focuses on a narrow set of logical/mathematical reasoning tasks. Exclusion of domains like code generation or long-context reasoning limits the generality of the findings.
>
> **Response:** We selected mathematical and logical reasoning tasks specifically because they offer deterministic, unique answers. This allows us to objectively evaluate the correctness of the final response and, by extension, the efficacy of the reasoning style. In domains like open-ended generation or complex code generation, evaluating the underlying "correctness" of a thought process is significantly more ambiguous. Our current scope allows us to unveil the performance characteristics of each style with high confidence.
>
>
> > The paper positions itself as an evaluation study rather than a contribution to machine learning methodology or theory. Without novel learning algorithms, model training, or theoretical analysis, it reads more like a large-scale empirical report than an ICLR-level research contribution.
>
> **Response:** Our work is submitted to the **Datasets and Benchmarks** track, where the goal is to provide high-quality resources and insights to the community. We analyze the behavior of different thinking styles across various tasks and scales, providing a necessary roadmap for selecting the optimal strategy for a given problem. We believe this empirical analysis is a critical contribution that aligns with the track's objectives, even without proposing a new architecture.
>
>
> > At line 393, the authors mentioned that they have fine-tuned a qwen-7b model, which failed to solve the challenge of reasoning path selection. What's the authors' opinions on a possible future solution? Is it possible to enable LLMs with this high-level capability via SFT (or RL)?
>
> **Response:** This failure case highlights that learning the optimal "thinking style" requires more than SFT, which often results in the model simply memorizing the "golden" reasoning path rather than internalizing the logic.
>
> While RL offers more potential than SFT, the main challenge is designing a dense reward function for the intermediate thinking steps. Currently, we rely heavily on sparse rewards (final answer correctness), which makes credit assignment difficult. Additionally, we observed that models can "hallucinate" thinking styles—mimicking the format of a style (e.g., ToT) without actually performing the logical operations required to solve the problem.
>
> For future work, we believe curriculum learning or contrastive learning offers a better path. For example, training the model to distinguish between a valid reasoning path and an invalid one (contrastive) might be more effective than simple generation tasks.

---

### Meta-Review · Area_Chair_ADW8 · 2026-01-08

**Summary:**

This paper proposes a benchmark (StyleBench) for evaluating multiple prompting-based reasoning styles across a range of LLMs. Reviewers agree that the work is clearly written, reproducible, and broad in empirical coverage. However, Reviewers also raise concens that the contribution is largely incremental since the benchmark repackages existing reasoning styles and datasets, and most findings confirm well-known trends such as scale effects and task–style alignment. Given these considerations, I recommend rejection.

**Reviewer Concerns:**

The authors clarify several design choices in experiment settings.

**Reviewer Scores:**

I believe no reviewers will change their rating.

---

### Decision · Program_Chairs · 2026-01-26

Reject